# NEUR2BILO: Neural Bilevel Optimization

**Justin Dumouchelle**[*]
University of Toronto

**Esther Julien**
TU Delft

**Jannis Kurtz**
University of Amsterdam

**Elias B. Khalil**
University of Toronto

## Abstract

Bilevel optimization deals with nested problems in which a *leader* takes the first decision to minimize their objective function while accounting for a *follower*'s best-response reaction. Constrained bilevel problems with integer variables are particularly notorious for their hardness. While exact solvers have been proposed for mixed-integer *linear* bilevel optimization, they tend to scale poorly with problem size and are hard to generalize to the non-linear case. On the other hand, problem-specific algorithms (exact and heuristic) are limited in scope. Under a data-driven setting in which similar instances of a bilevel problem are solved routinely, our proposed framework, NEUR2BILO, embeds a neural network approximation of the leader's or follower's value function, trained via supervised regression, into an easy-to-solve mixed-integer program. NEUR2BILO serves as a heuristic that produces high-quality solutions extremely fast for four applications with linear and non-linear objectives and pure and mixed-integer variables.

## 1  Introduction

**A motivating application.**  Consider the following *discrete network design problem* (DNDP) [47, 48]. A transportation planning authority seeks to minimize the average travel time on a road network represented by a directed graph of nodes $N$ and links $A_1$ by investing in constructing a set of roads (i.e., links) from a set of options $A_2$, subject to a budget $B$. The planner knows the number of vehicles that travel between any origin-destination (O-D) pair of nodes. A good selection of links should take into account the drivers' reactions to this decision. One common assumption is that drivers will optimize their O-D paths such that a *user equilibrium* is reached. This is known as *Wardrop's second principle* in the traffic assignment literature, an equilibrium in which "no driver can unilaterally reduce their travel costs by shifting to another route" [41]. This is in contrast to the *system optimum*, an equilibrium in which a central planner dictates each driver's route, an unrealistic assumption that would not require bilevel modeling. A link cost function is used to model the travel time on an edge as a function of traffic. Let $c_{ij} \in \mathbb{R}_+$ be the capacity (vehicles per hour (vph)) of a link and $T_{ij} \in \mathbb{R}_+$ the free-flow travel time (i.e., travel time on the link without congestion). The US Bureau of Public Roads uses the following widely accepted formula to model the travel time $t(y_{ij})$ on a link used by $y_{ij}$ vehicles per hour: $t(y_{ij}) = T_{ij}(1 + 0.15(y_{ij}/c_{ij})^4)$. As the traffic $y_{ij}$ grows to exceed the capacity $c_{ij}$, a large quartic increase in travel time is incurred [41].

*Bilevel optimization* (BiLO) [4] models the DNDP and many problems in which an agent (the *leader*) makes decisions that minimize their cost function subject to another agent's (the *follower*'s) best response. In the DNDP, the *leader* is the transportation planner and the *follower* is the population of

---

[*]Corresponding author: `justin.dumouchelle@mail.utoronto.ca`

38th Conference on Neural Information Processing Systems (NeurIPS 2024).

drivers, giving rise to the following optimization problem

$$\min_{\mathbf{x}\in\{0,1\}^{|A_2|},\mathbf{y}} \quad \sum_{(i,j)\in A} y_{ij} t(y_{ij})$$

$$\text{s.t.} \quad \sum_{(i,j)\in A_2} g_{ij} x_{ij} \leq B,$$

$$\mathbf{y} \in \arg\min_{\mathbf{y}'\in\mathbb{R}_+^{|A|}} \quad \sum_{(i,j)\in A} \int_0^{y'_{ij}} t_{ij}(v)dv$$

$$\text{s.t.} \quad \mathbf{y}' \text{ is a valid network flow,}$$

$$x_{ij} = 0 \implies y'_{ij} = 0,$$

where $A_2 \cap A_1 = \emptyset, A = A_1 \cup A_2$. The leader minimizes the total travel time across all links subject to a budget constraint and the followers' equilibrium which is expressed as a network flow on the graph augmented by the leader's selected edges that satisfies O-D demands; the integral in the follower's objective models the desired equilibrium and evaluates to $T_{ij}y'_{ij} + \frac{0.15T_{ij}}{5c_{ij}^4}\left(y'_{ij}\right)^5$.

Going beyond the DNDP, Dempe [16] lists more than 70 applications of BiLO ranging from pricing in electricity markets (leader is an electricity-supplying retailer that sets the price to maximize profit, followers are consumers who react accordingly to satisfy their demands [60]) to interdiction problems in security settings (leader inspects a budgeted subset of nodes on a road network, follower selects a path such that they evade inspection [56]).

**Scope of this work.** We are interested in *mixed-integer non-linear bilevel optimization* problems, simply referred to hereafter as *bilevel optimization* or BiLO, a very general class of bilevel problems where all constraints and objectives may involve non-linear terms and integer variables. At a high level, we have identified three limitations of existing computational methods for BiLO:

1. The state-of-the-art exact solvers of Fischetti et al. [24] and Tahernejad et al. [53] are limited to mixed-integer bilevel *linear* problems and do not scale well. When high-quality solutions to large-scale problems are sought after, such exact solvers may be prohibitively slow.
2. Specialized algorithms, heuristic or exact, do not generalize beyond the single problem they were designed for. For instance, the state-of-the-art exact Knapsack Interdiction solver [57] only works for a single knapsack constraint and fails with two or more, a significant limitation even if one is strictly interested in knapsack-type problems.
3. Existing methods, exact or heuristic, generic or specialized, are not designed for the "data-driven algorithm design" setting [3] in which similar instances are routinely solved and the goal is to construct generalizable high-efficiency algorithms that leverage historical data.

NEUR2BILO (for *Neural Bilevel Optimization*) is a learning-based framework for bilevel optimization that deals with these issues simultaneously. The following observations make NEUR2BILO possible:

1. **Data collection is "easy":** For a fixed decision of the leader's, the optimal value of the follower can be computed by an appropriate (single-level) solver (e.g., for mixed-integer programming (MIP) or convex programming), enabling the collection of samples of the form: (leader's decision, follower's value, leader's value).
2. **Offline learning in the data-driven setting:** While obtaining data online may be prohibitive, access to historical training instances affords us the ability to construct, offline, a large dataset of samples that can then serve as the basis for learning an approximate value function using supervised regression. The output of this training is a regressor mapping a pair consisting of an instance and a leader's decision to an estimated follower or leader value.
3. **MIP embeddings of neural networks:** If the regressor is MIP-representable, e.g., a feedforward ReLU neural network or a decision tree, it is possible to use a MIP solver to find the leader's decision that minimizes the regressor's output. This MIP, which includes any leader constraints, thus serves as an approximate single-level surrogate of the original bilevel problem instance.
4. **Follower constraints via the value function reformulation:** The final ingredient of the NEUR2BILO recipe is to include any of the follower's constraints, some of which may

involve leader variables. This makes the surrogate problem a heuristic version of the well-known *value function reformulation* (VFR) in BiLO. The VFR transforms a bilevel problem into a single-level one, assuming that one can represent the follower's value (as a function of the leader's decision) compactly. This is typically impossible as the value function may require an exponential number of constraints, a bottleneck that is circumvented by our small (approximate) regression models.

5. **Theoretical guarantees:** For interdiction problems, a class of BiLO problems that attract much attention, NEUR2BILO solutions have a constant, additive absolute optimality gap which mainly depends on the prediction accuracy of the regression model.

Through a series of experiments on (i) the bilevel knapsack interdiction problem, (ii) the "critical node problem" from network security, (iii) a donor-recipient healthcare problem, and (iv) the DNDP, we will show that NEUR2BILO is easy to train and produces, very quickly, heuristic solutions that are competitive with state-of-the-art methods.

## 2 Background

Bilevel optimization (BiLO) deals with hierarchical problems where the *leader* (or *upper-level*) problem decides on $\mathbf{x} \in \mathcal{X}$ and parameterizes the *follower* (or *lower-level*) problem that decides on $\mathbf{y} \in \mathcal{Y}$; the sets $\mathcal{X}$ and $\mathcal{Y}$ represent the domains of the variables (continuous, mixed-integer, or pure integer). Both problems have their own objectives and constraints, resulting in the following model:

$$\min_{\mathbf{x} \in \mathcal{X}, \mathbf{y}} \quad F(\mathbf{x}, \mathbf{y}) \tag{1a}$$

$$\text{s.t.} \quad G(\mathbf{x}, \mathbf{y}) \geq \mathbf{0}, \tag{1b}$$

$$\mathbf{y} \in \arg\max_{\mathbf{y}' \in \mathcal{Y}} \{ f(\mathbf{x}, \mathbf{y}') : g(\mathbf{x}, \mathbf{y}') \geq \mathbf{0} \}, \tag{1c}$$

where we consider the general mixed-integer non-linear case with $F, f : \mathcal{X} \times \mathcal{Y} \to \mathbb{R}$, $G : \mathcal{X} \times \mathcal{Y} \to \mathbb{R}^{m_1}$, and $g : \mathcal{X} \times \mathcal{Y} \to \mathbb{R}^{m_2}$ non-linear functions of the upper-level $\mathbf{x}$ and lower-level variables $\mathbf{y}$.

The applicability of exact (i.e., global) approaches critically depends on the nature of the lower-level problem. A continuous lower-level problem admits a single-level reformulation that leverages the Karush-Kuhn-Tucker (KKT) conditions as constraints on $\mathbf{y}$. For linear programs in the lower level, strong duality conditions can be used in the same way. Solving a BiLO problem with integers in the lower level necessitates more sophisticated methods such as branch and cut [17, 24] along with some assumptions: DeNegre and Ralphs [17] do not allow for coupling constraints (i.e., $G(\mathbf{x}, \mathbf{y}) = G(\mathbf{x})$) and both methods do not allow continuous upper-level variables to appear in the linking constraints ($g(\mathbf{x}, \mathbf{y})$). Other approaches, such as Benders decomposition, are also applicable [25]. Gümüş and Floudas [29] propose single-level reformulations of mixed-integer non-linear BiLO problems using polyhedral theory, an approach that only works for small problems. Later, "branch-and-sandwich" methods were proposed [33, 45] where bounds on both levels' value functions are used to compute an optimal solution. Algorithms for non-linear BiLO generally do not scale well. Kleinert et al. [32] survey more exact methods.

**Assumptions.** In what follows, we make the following standard assumptions:

1. Either (i) the follower's problem has a feasible solution for each $\mathbf{x} \in \mathcal{X}$, or (ii) there are no coupling constraints in the leader's problem, i.e., $G(\mathbf{x}, \mathbf{y}) = G(\mathbf{x})$;
2. The optimal follower value is always attained by a feasible solution [see 5, Section 7.2].

**Value function reformulation.** We consider the so-called *optimistic* setting: if the follower has multiple optima for a given decision of the leader's, the one that optimizes the leader's objective is implemented. We can then rewrite problem (1) using the *value function reformulation* (VFR):

$$\min_{\mathbf{x} \in \mathcal{X}, \mathbf{y} \in \mathcal{Y}} \quad F(\mathbf{x}, \mathbf{y}) \tag{2a}$$

$$\text{s.t.} \quad G(\mathbf{x}, \mathbf{y}) \geq \mathbf{0}, \tag{2b}$$

$$g(\mathbf{x}, \mathbf{y}) \geq \mathbf{0}, \tag{2c}$$

$$f(\mathbf{x}, \mathbf{y}) \geq \Phi(\mathbf{x}), \tag{2d}$$

with the *optimal lower-level value function* defined as

$$\Phi(\mathbf{x}) = \max_{\mathbf{y} \in \mathcal{Y}} \{ f(\mathbf{x}, \mathbf{y}) : g(\mathbf{x}, \mathbf{y}) \geq \mathbf{0} \}. \tag{3}$$

Lozano and Smith [39] used this formulation to construct an exact algorithm (without any public code) for solving mixed-integer non-linear BiLO problems with purely integer upper-level variables. Sinha et al. [50, 51, 52] propose a family of evolutionary heuristics for continuous non-linear BiLO problems that approximate the optimal value function by using quadratic and Kriging (i.e., a function interpolation method) approximations. Taking it one step further, Beykal et al. [9] extend the framework of the previous authors to handle mixed-integer variables in the lower level.

# 3 Methodology

NEUR2BILO refers to two learning-based single-level reformulations for general BiLO problems. The reformulations rely on representing the thorny nested structure of a BiLO problem with a trained regression model that predicts either the upper-level or lower-level value functions. Appendix B includes pseudocode for data collection, training, and model deployment.

## 3.1 NEUR2BILO

**Upper-level approximation.** The obvious bottleneck in solving BiLO problems is their nested structure. One rather straightforward way of circumventing this difficulty is to get rid of the lower level altogether in the formulation, but predict its optimal value. Namely, we predict the optimal upper-level objective value function as

$$\text{NN}^u(\mathbf{x}; \Theta) \approx F(\mathbf{x}, \mathbf{y}^\star), \tag{4}$$

where $\Theta$ are the weights of a neural network, $F$ the objective function of the leader (2b), and $\mathbf{y}^\star$ an optimal solution to the lower level problem (3). To train such a model, one can sample $\mathbf{x}$ from $\mathcal{X}$, solve (3) to obtain an optimal lower-level solution $\mathbf{y}^\star$, and subsequently compute a label $F(\mathbf{x}, \mathbf{y}^\star)$. We can then model the single-level problem as

$$\min_{\mathbf{x} \in \mathcal{X}} \quad \text{NN}^u(\mathbf{x}; \Theta) \quad \text{s.t. } G(\mathbf{x}) \geq \mathbf{0}, \tag{5}$$

where we only optimize for $\mathbf{x}$ and thus dismiss the lower-level constraints and objective function. A trained feedforward neural network $\text{NN}^u(\cdot; \Theta)$ with ReLU activations can be represented as a mixed-integer linear program (MILP) [22], where now the input (and output) of the network are decision variables. With this representation, Problem (5) becomes a single-level problem and can be solved using an off-the-shelf MIP solver. Note that linear and decision tree-based models also admit MILP representations [38].

This reformulation is similar to the approach by Bagloee et al. [2], wherein the upper-level value function is predicted using linear regression. Our method differs in that it is not iterative and does not require the use of "no-good cuts" (which avoid reappearing solutions $\mathbf{x}$). As such, our method is extremely efficient as will be shown experimentally.

The formulation of (5) only allows for problem classes that do not have coupling constraints, i.e., $G(\mathbf{x}, \mathbf{y}) = G(\mathbf{x})$. Moreover, the feasibility of a solution $\mathbf{x}$ in the original BiLO problem is not guaranteed, an issue that will be addressed later in this section (see **Bilevel feasibility.**).

**Lower-level approximation.** This method makes use of the VFR (2). The VFR moves the nested complexity of a BiLO to constraint (2d), where the right-hand side is the optimal value of the lower-level problem, parameterized by $\mathbf{x}$. We introduce a learning-based VFR in which $\Phi(\mathbf{x})$ is approximated by a regression model with parameters $\Theta$:

$$\text{NN}^l(\mathbf{x}; \Theta) \approx \Phi(\mathbf{x}). \tag{6}$$

Both $\text{NN}^l$ and $\text{NN}^u$ take in a leader's decision as input and require solving the follower (3) for data generation. By replacing $\Phi(\mathbf{x})$ with $\text{NN}^l(\mathbf{x}; \Theta)$ in (2d) and introducing a slack variable $s \in \mathbb{R}_+$, the

surrogate VFR reads as:

$$\min_{\substack{\mathbf{x} \in \mathcal{X}, \mathbf{y} \in \mathcal{Y} \\ s \geq 0}} \quad F(\mathbf{x}, \mathbf{y}) + \lambda s \tag{7a}$$

$$\text{s.t.} \quad G(\mathbf{x}, \mathbf{y}) \geq \mathbf{0}, \tag{7b}$$

$$g(\mathbf{x}, \mathbf{y}) \geq \mathbf{0}, \tag{7c}$$

$$f(\mathbf{x}, \mathbf{y}) \geq \text{NN}^l(\mathbf{x}; \Theta) - s. \tag{7d}$$

All follower and leader constraints of the original BiLO problem are part of Problem (7). However, without the slack variable $s$, the problem could become infeasible due to inaccurate predictions by the neural network. This happens when $\text{NN}^l(\mathbf{x}; \Theta)$ strictly overestimates the follower's optimal value for each $\mathbf{x}$. In this case, there does not exist a follower decision for which Constraint (7d) is satisfied. A value of $s > 0$ can be used to make Constraint (7d) satisfiable at a cost of $\lambda s$ in the objective, guaranteeing feasibility.

**Bilevel feasibility.** Given a solution $\mathbf{x}^\star$ or a solution pair $(\mathbf{x}^\star, \tilde{\mathbf{y}})$ returned by our upper- or lower-level approximations, respectively, we would like to produce a lower-level solution $\mathbf{y}^\star$ such that $(\mathbf{x}^\star, \mathbf{y}^\star)$ is bilevel-feasible, i.e., it satisfies the original BiLO in (1). The following procedure achieves this goal:

1. Compute the follower's optimal value under $\mathbf{x}^\star$, $\Phi(\mathbf{x}^\star)$, by solving (3).
2. Compute a bilevel-feasible follower solution $\mathbf{y}^\star$ by solving problem (2) with fixed $\mathbf{x}^\star$ and the right-hand side of (2d) set to $\Phi(\mathbf{x}^\star)$, a constant. Return $(\mathbf{x}^\star, \mathbf{y}^\star)$.

If only Assumption 1(i) is satisfied, then only the lower-level approximation is applicable and this procedure guarantees an optimistic bilevel-feasible solution for it. If only Assumption 1(ii) is satisfied, then this procedure can detect in Step 1 that an upper-level approximation's solution $\mathbf{x}^\star$ does not admit a follower solution, i.e., that it is infeasible, or calculates a feasible $\mathbf{y}^\star$ if one exists in Step 2. If both Assumptions 1(i) and 1(ii) are satisfied simultaneously, then this procedure guarantees an optimistic bilevel-feasible solution for either approximation.

**Upper- v.s. lower-level level approximation.** Here, we note two important trade-offs between the upper- and lower-level approximations.

- **Generality**: Example C.1 in Appendix C shows that under Assumption 1(ii), it may happen that solving the upper-level approximation problem variant (5) returns an infeasible solution while the lower-level variant (7) does not.
- **Scalability**: The upper-level approximation has fewer variables and constraints than its lower-level counterpart as it does not represent the follower's problem directly. For problems in which the lower-level problem is large, e.g., necessitating constraints for each node and link to enforce a network flow in the follower solution as in the DNDP from the introduction, this property makes the upper-level approximation easier to solve, possibly at a sacrifice in final solution quality. This tradeoff will be assessed experimentally.

**Limitations.** Since NEUR2BILO is in essence a learning-based *heuristic*, it does not guarantee an optimal solution to the bilevel problem. However, it guarantees a feasible solution with the lower-level approximation and can only give an infeasible solution while using the upper-level approximation when only Assumption 1(ii) is satisfied. Moreover, as will be shown in Section 3.3, the performance of NEUR2BILO depends on the regression error, which is generally the case when integrating machine learning in optimization algorithms. Empirically, we note that the prediction error achieved on every problem is very low (see Appendix K.3).

## 3.2 Model architecture

For ease of notation in previous sections, all regression models take as input the upper-level decision variables. However, in our experiments, we leverage instance information as well to train *a single model* that can be deployed on a family of instances. This is done by leveraging information such as coefficients in the objective and constraints for each problem.

For the model's architecture, the general principle deployed is to first explicitly represent or learn instance-based features. The second is to combine instance-based features with (leader) decision variable information to make predictions.

The overall architecture can be summarized as the following set of operations. Fix a particular instance of a BiLO problem and let $n$ be the number of leader variables, $\mathbf{f}_i$ a vector of features for each leader variable $\mathbf{x}_i$ (independently of the variable's value), and $h(\mathbf{x}_i)$ a feature map that describes the $i$th leader variable for a specific value of that variable. The functions $\Psi^s$, $\Psi^d$, and $\Psi^v$ are neural networks with appropriate input-output dimensions. The vector $\Theta$ includes all learnable parameters of networks $\Psi^s$, $\Psi^d$, and $\Psi^v$. The functions SUM, CONCAT, and AGGREGATE sum up a set of vectors, concatenate two vectors into a single column vector, and aggregate a set of scalar values (e.g., by another neural network or simply summing them up), respectively. Our final objective value predictions are then given by the following sequence of steps:

1. Embedding the set of variable features $\{\mathbf{f}_i\}$ using a set-based architecture, e.g., the same network $\Psi^d$, summing up the resulting $n$ variable embeddings, then passing the resulting vector to network $\Psi^s$, yielding a vector we refer to as the INSTANCEEMBEDDING:
$$\text{INSTANCEEMBEDDING} = \Psi^s(\text{SUM}(\{\Psi^d(\mathbf{f}_i)\}_{i=1}^n)).$$
This is akin to the DeepSets approach of Zaheer et al. [58]. However, note that this step can alternatively be done via a feedforward or graph neural network depending on the problem structure.

2. Conditional on a specific assignment of values to the leader's decision vector $\mathbf{x}$, a per-variable embedding is computed by network $\Psi^v$ to allow for interactions between the INSTANCEEMBEDDING and the specific assignment of variable $i$ as represented by $h(\mathbf{x}_i)$:
$$\text{VARIABLEEMBEDDING}(i) = \Psi^v(\text{CONCAT}(h(\mathbf{x}_i), \text{INSTANCEEMBEDDING})).$$

3. The final value prediction for either of our approximations aggregates the variable embeddings possibly after passing them through a function $g_i$:
$$\text{NN}(\mathbf{x}; \Theta) = \text{AGGREGATE}(\{g_i(\text{VARIABLEEMBEDDING}(i))\}_{i=1}^n).$$
For example, if the follower's objective is a linear function and VARIABLEEMBEDDING$(i)$ is a scalar, then it is useful to use the variable's known objective function coefficient $d_i$ here, i.e.: $g_i(\text{VARIABLEEMBEDDING}(i)) = d_i \cdot \text{VARIABLEEMBEDDING}(i)$. The final step is to aggregate the per-variable $g_i(\cdot)$ outputs, e.g., by a summation for linear or separable objective functions.

NEUR2BILO is largely agnostic to the learning model utilized as long as it is MILP-representable. In our experiments, we primarily focus on neural networks, but for some problems also explore the use of gradient-boosted trees. More details on the specific architectures for each problem can be found in Appendix K.1.

### 3.3 Approximation guarantees

**Lower-level approximation.** Next, we present an approximation guarantee for the lower-level approximation with $\text{NN}^l(\mathbf{x}; \Theta)$. Appendix D includes the complete proofs.

Since the prediction of the neural network is only an approximation of the true optimal value of the follower's problem $\Phi(\mathbf{x})$, NEUR2BILO may return sub-optimal solutions for the original problem (1). We derive approximation guarantees for a specific setup that appears in interdiction problems: the leader and the follower have the same objective function (i.e., $f(\mathbf{x}, \mathbf{y}) = F(\mathbf{x}, \mathbf{y})$ for all $\mathbf{x} \in \mathcal{X}, \mathbf{y} \in \mathcal{Y}$), and Assumption 1(i) holds. Consider a neural network that approximates the optimal value of the follower's problem up to an absolute error of $\alpha > 0$, i.e.,
$$|\text{NN}^l(\mathbf{x}; \Theta) - \Phi(\mathbf{x})| \leq \alpha \quad \text{for all } \mathbf{x} \in \mathcal{X}. \tag{8}$$
Furthermore, we define the parameter $\Delta$ as the maximum difference $f(\mathbf{x}, \mathbf{y}) - f(\mathbf{x}, \mathbf{y}') \geq 0$ over all $\mathbf{x} \in \mathcal{X}, \mathbf{y}, \mathbf{y}' \in \mathcal{Y}$ such that no $\tilde{\mathbf{y}} \in \mathcal{Y}$ exists which has function value $f(\mathbf{x}, \mathbf{y}) > f(\mathbf{x}, \tilde{\mathbf{y}}) > f(\mathbf{x}, \mathbf{y}')$. We can bound the approximation guarantee of the lower-level NEUR2BILO as follows:

**Theorem 3.1.** *If the leader and the follower have the same objective function and $\lambda > 1$,* NEUR2BILO *returns a feasible solution $(\mathbf{x}^\star, \mathbf{y}^\star)$ for Problem* (1) *with objective value*
$$f(\mathbf{x}^\star, \mathbf{y}^\star) \leq \text{opt} + 3\alpha + \frac{2}{\lambda}\Delta,$$
*where* opt *is the optimal value of* (1) *and $\lambda$ the penalty term in* (7a) *.*

**Upper-level approximation.** As Example C.1 shows, it may happen that the upper-level surrogate problem (5) returns an infeasible solution and hence no approximation guarantee can be derived in this case. However, in the case where all leader solutions are feasible and the neural network predicts for every $\mathbf{x} \in \mathcal{X}$ an upper-level objective value that deviates at most $\alpha > 0$ from the true one, then the returned solution trivially approximates the true optimal value with an absolute error of at most $2\alpha$. This follows since the worst that can happen is that the objective value of the optimal solution is overestimated by $\alpha$ while a solution with objective value opt $+ 2\alpha$ is underestimated by $\alpha$ and hence has the same predicted value as the optimal solution. Problem (5) then may return the latter sub-optimal solution.

## 4 Experimental Setup

**Benchmark problems** and their characteristics are summarized in Table 1; their MIP formulations are deferred to Appendix E and brief descriptions follow:

- **Knapsack interdiction (KIP) [10]:** The leader interdicts a subset of at most $k$ items and the follower solves a knapsack problem over the remaining (non-interdicted) items. The leader aims to minimize the follower's (maximization) objective.
- **Critical node problem (CNP) [18, 11]:** This problem regards the protection (by the leader) of resources in a network against malicious follower attacks. It has applications in the protection of computer networks against cyberattacks as demonstrated by Dragotto et al. [18].
- **Donor-recipient problem (DRP) [27]:** This problem relates to the donations given by certain agencies to countries in need of, e.g., healthcare projects. The leader (the donor agency) decides on which proportion of the cost, per project, to subsidize, whereas the follower (a country) decides which projects it implements.
- **Discrete network design problem (DNDP) [47]:** This is the problem described in Section 1. We build on the work of Rey [47] who provided benchmark instances for the transportation network of Sioux Falls, South Dakota, and an implementation of the state-of-the-art method of Fontaine and Minner [25]. This network and corresponding instances are representative of the state of the DNDP in the literature.

| | | | Leader | | | Follower | |
|---|---|---|---|---|---|---|---|
| Problem | | x | Obj. | Cons. | y | Obj. | Cons. |
| KIP | ($\downarrow\uparrow$) | B | Lin | Lin | B | Lin | Lin |
| CNP | ($\uparrow\uparrow$) | B | BLin | Lin | B | BLin | Lin |
| DRP | ($\uparrow\uparrow$) | C | Lin | Lin | MI | Lin | BLin |
| DNDP | ($\downarrow\uparrow$) | B | NLin | Lin | C | NLin | Lin |

Table 1: Problem class characteristics. All problems have a single budget constraint in the leader; for the follower, the DNDP has network flow constraints whereas other problems have a knapsack constraint. The arrows refer to minimization ($\downarrow$) or maximization ($\uparrow$) in leader and follower, respectively. B = Binary, C = Continuous, MI = Mixed-Integer, Lin = Linear, BLin = Bilinear, NLin = Non-Linear.

**Baselines.** As mentioned previously, the branch-and-cut (B&C) algorithm by Fischetti et al. [24] is considered to be state-of-the-art for solving mixed-integer linear BiLO. The method is applicable if the continuous variables of the leader do not appear in the follower's constraints. Both KIP and CNP meet these assumptions. This algorithm will act as the baseline for these problems. For DRP, we compare against the results produced by an algorithm in the branch-and-cut paradigm (B&C+) from Ghatkar et al. [27]. For DNDP, the follower's problem only has continuous variables, so the baseline is a method based on KKT conditions (MKKT) [25]. Of the learning-based approaches for BiLO, we compare against Zhou et al. [59], given the generality of their approach and the availability of source code. NEUR2BILO decisively outperforms this method on KIP, finding solutions with 10-100$\times$ smaller mean relative error roughly 1000$\times$ faster; full results are deferred to Appendix F.

**Data collection & Training.** For each problem class, data is collected by sampling feasible leader decisions $\mathbf{x}$ and then solving $\Phi(\mathbf{x})$ to compute either the upper- or lower-level objectives as labels. We then train regression models to minimize the least-squares error on the training samples. Typically, data collection and training take less than one hour, a negligible cost given that for larger instances

baseline methods require more time *per instance*. Additionally, the same trained model can be used on multiple unseen test instances. We report times for data collection and training in Appendix K.2.

**Evaluation & Setup.** For evaluation in KIP, CNP, and DRP, all solving was limited to 1 hour. For DNDP, we consider a more limited-time regime, wherein we compare NEUR2BILO at 5 seconds against the baseline at 5, 10, and 30 seconds. For all problems, we evaluate both the lower- and upper-level approximations with neural networks, namely $NN^l$ and $NN^u$, respectively. For $NN^l$ we set $\lambda = 1$ for all results presented in the main paper. Details of the computing setup are provided in Appendix J. Our code and data are available at `https://github.com/khalil-research/Neur2BiLO`.

## 5 Experimental Results

We summarize the results as measured by average solution times and mean relative errors (MREs). The relative error on a given instance is computed as $100 \cdot \frac{|obj_\mathcal{A} - obj_{best}|}{|obj_{best}|}$, where $obj_\mathcal{A}$ is the value of the solution found by method $\mathcal{A}$ and $obj_{best}$ is the best-known objective value for that instance. These results are reported in Table 2. More detailed results and box-plots of the distributions of relative errors are in Appendices G and H. Our experimental design answers the following questions:

**Q1: Can NEUR2BILO find high-quality solutions quickly on classical interdiction problems?**
Table 2 compares NEUR2BILO to the B&C algorithm of Fischetti et al. [24]. NEUR2BILO terminates in 1-2% of the time required by B&C on the smaller ($n \leq 30$) well-studied KIP instances of Tang et al. [54]. However, when the instance size increases to $n = 100$, both $NN^l$ and $NN^u$ find much better solutions than NEUR2BILO in roughly 30 seconds, even when B&C runs for the full hour. Furthermore, Table 4 in Appendix G shows that B&C requires 10 to $1,000\times$ more time than $NN^l$ or $NN^u$ to find equally good solutions. In addition, the best solutions found by B&C at the termination times of $NN^l$ or $NN^u$ are generally worse, even for small instances.

**Q2: Do these computational results extend to non-linear and more challenging BiLO problems?**
Interdiction problems such as the KIP are well-studied but are only a small subset of BiLO. We will shift attention to the more practical problems, starting with the CNP (Table 2). CNP includes terms that are bilinear (i.e., $z = xy$) in the upper- and lower-level variables, resulting in a much more challenging problem for general-purpose B&C. In this case, both $NN^l$ and $NN^u$ tend to outperform B&C as the problem size increases. In addition, the results on incumbents reported in Table 5 in Appendix G are as good, if not even stronger than those of KIP.

Secondly, we discuss DRP (Table 6 in Appendix G). For DRP, we evaluate on the most challenging instances from Ghahtarani et al. [26], all of which have gaps of $\sim 50\%$ at a 1-hour time limit with B&C+, a specialized B&C-based algorithm. Here $NN^u$ performs remarkably well: it finds the best-known solutions on every single instance in roughly $\sim 0.1$ seconds at an average improvement in solution quality of 26% over B&C+.

**Q3: How does NEUR2BILO perform on BiLO problems with complex constraints?** Given that NEUR2BILO has strong performance on benchmarks with budget constraints, the next obvious question is whether it can be applied to BiLO problems that have complex constraints. To answer this, we will refer to the results in Table 2 for the DNDP. In this setting, we focus on a limited-time regime wherein we compare NEUR2BILO with a 5-second time limit to MKKT at time limits 5, 10, and 30 seconds. $NN^u$ can achieve high-quality solutions much faster than any other method with only a minor sacrifice in solution quality, making it a great candidate for domains where interactive decision-making is needed (e.g., what-if analysis of various candidate roads, budgets, etc.).

$NN^l$, on the other hand, takes longer than $NN^u$ but computes solutions that are more competitive with the baseline, the latter requiring $5\times$ more time. We suspect that the better solution quality from $NN^l$ is due to its explicit modeling of feasible lower-level decisions that "align" with the predictions, whereas $NN^u$ may simply extrapolate poorly. In terms of computing time, one computational burden for $NN^l$ is the requirement to model the non-linear upper- and lower-level objectives, which requires a piece-wise linear approximation based on Fontaine and Minner [25], a step that introduces additional variables and constraints. Appendix G includes results for DNDP with gradient-boosted trees (GBT), demonstrating that other learning models are directly applicable and, in some cases, may even lead to better solution quality, faster optimization, and simpler implementation.

| | | Knapsack Interdiction Problem | | | | | | | |
|---|---|---|---|---|---|---|---|---|---|
| # Items | Interdiction | $\mathrm{NN}^l$ | | $\mathrm{NN}^u$ | | G-VFA | | B&C | |
| $(n)$ | Budget $(k)$ | MRE | Time | MRE | Time | MRE | Time | MRE | Time |
| 18 | 5 | 1.48 | 0.59 | 1.48 | 0.34 | 1.82 | **0.14** | **0.00** | 9.55 |
| 18 | 9 | 1.51 | 0.59 | 1.51 | 0.43 | 3.97 | **0.22** | **0.00** | 5.81 |
| 18 | 14 | **0.00** | 0.22 | **0.00** | 0.17 | 64.22 | **0.03** | **0.00** | 0.39 |
| 20 | 5 | 0.41 | 0.62 | 0.41 | 0.45 | 2.19 | **0.25** | **0.00** | 23.18 |
| 20 | 10 | 0.99 | 0.66 | 0.99 | 0.58 | 0.99 | **0.36** | **0.00** | 10.27 |
| 20 | 15 | 3.57 | 0.32 | 3.57 | 0.19 | 23.39 | **0.02** | **0.00** | 0.94 |
| 22 | 6 | 0.71 | 0.19 | 0.71 | **0.18** | 0.42 | 0.18 | **0.00** | 42.30 |
| 22 | 11 | 1.01 | 0.28 | 1.01 | **0.28** | 1.08 | 0.33 | **0.00** | 16.26 |
| 22 | 17 | 14.43 | 0.24 | 14.43 | 0.15 | 14.43 | **0.13** | **0.00** | 0.68 |
| 25 | 7 | 0.44 | 2.66 | 0.44 | 2.42 | 0.44 | **0.64** | **0.00** | 137.96 |
| 25 | 13 | 1.42 | 2.75 | 1.42 | 2.79 | 3.85 | **1.24** | **0.00** | 48.43 |
| 25 | 19 | 2.49 | 0.48 | 2.49 | 0.38 | 2.49 | **0.13** | **0.00** | 1.77 |
| 28 | 7 | 0.39 | 0.67 | 0.39 | 0.74 | 0.26 | **0.62** | **0.00** | 309.18 |
| 28 | 14 | 0.75 | 2.10 | 0.75 | 1.45 | 1.37 | **1.29** | **0.00** | 120.74 |
| 28 | 21 | 1.14 | 0.45 | 1.14 | 0.49 | 3.16 | **0.31** | **0.00** | 4.92 |
| 30 | 8 | **0.00** | 1.54 | **0.00** | 1.54 | 0.43 | **0.97** | **0.00** | 792.44 |
| 30 | 15 | 0.49 | 3.64 | 0.49 | 3.06 | 0.75 | **1.35** | **0.00** | 187.23 |
| 30 | 23 | 2.29 | 1.08 | 2.29 | 0.73 | 4.48 | **0.25** | **0.00** | 5.65 |
| 100 | 25 | 0.93 | 10.02 | 0.93 | 8.40 | **0.00** | **4.19** | 8.09 | 3,600.40 |
| 100 | 50 | 0.96 | 51.68 | 0.96 | 49.28 | 0.04 | 53.74 | 8.96 | 3,600.44 |
| 100 | 75 | **0.08** | 24.69 | **0.08** | 23.78 | 0.12 | 35.27 | 5.87 | 3,600.52 |
| Avg. $n \leq 30$ | | 1.86 | 1.06 | 1.86 | 0.91 | 7.21 | **0.47** | **0.00** | 95.43 |
| Avg. $n = 100$ | | 0.66 | 28.80 | 0.66 | **27.15** | **0.05** | 31.07 | 7.64 | 3,600.45 |

| | Critical Node Problem | | | | | |
|---|---|---|---|---|---|---|
| # Nodes | $\mathrm{NN}^l$ | | $\mathrm{NN}^u$ | | B&C | |
| $(|V|)$ | MRE | Time | MRE | Time | MRE | Time |
| 10 | 3.20 | 0.04 | 2.75 | **0.02** | **1.01** | 4.24 |
| 25 | 2.60 | 0.23 | 1.77 | **0.05** | **0.73** | 3,244.20 |
| 50 | 1.42 | 0.38 | 0.98 | **0.10** | **0.67** | 3,600.30 |
| 100 | 1.12 | 0.48 | **0.56** | **0.42** | 1.79 | 3,600.65 |
| 300 | 2.01 | 1.12 | **0.33** | **0.83** | 2.32 | 3,600.54 |
| 500 | 1.33 | 1.69 | **0.45** | **1.19** | 2.47 | 3,600.80 |
| Average | 1.95 | 0.66 | **1.14** | **0.43** | 1.50 | 2,941.79 |

| | | Discrete Network Design Problem | | | | | | |
|---|---|---|---|---|---|---|---|---|
| | | $\mathrm{NN}^l$ | | $\mathrm{NN}^u$ | | | MKKT | |
| # Edges | Budget | MRE | Time | MRE | Time | MRE-5 | MRE-10 | MRE-30 |
| 10 | 0.25 | 1.21 | 2.95 | 0.36 | **0.01** | 5.78 | 0.51 | **0.10** |
| 10 | 0.5 | 0.73 | 3.35 | 1.22 | **0.01** | 6.47 | 2.17 | **0.00** |
| 10 | 0.75 | 0.47 | 2.80 | 1.32 | **0.00** | 5.80 | **0.02** | 0.06 |
| 20 | 0.25 | 6.05 | 5.02 | 2.64 | **0.02** | 7.78 | 5.12 | **0.85** |
| 20 | 0.5 | 1.01 | 4.91 | 4.36 | **0.03** | 6.00 | 2.52 | **0.64** |
| 20 | 0.75 | 0.85 | 4.47 | 0.91 | **0.01** | 7.87 | 0.22 | **0.11** |
| Average | | 1.72 | 3.92 | 1.80 | **0.01** | 6.62 | 1.76 | **0.29** |

Table 2: Mean relative error (MRE) and solving times for KIP, CNP, and DNDP. For KIP with $n \leq 30$, we directly evaluate on the 180 instances (10 per size) of Tang et al. [54]; each value is the average over 10 instances. For $n = 100$, our evaluation instances (100 per size) are generated using the same procedure of Tang et al. [54]. The no-learning baseline G-VFA is a VFR using the follower's greedy solution as lower-level value function approximation. For CNP, each row is averaged over 300 instances that are randomly sampled using the procedure described in Dragotto et al. [18]. For DNDP, each row is averaged over 10 instances from Rey [47]. The budget is a fraction of the total cost of all 30 possible candidate links; see Appendix K.2 for more details.

**Q4: Can approximations derived from heuristics be useful?** We now refer back to KIP and focus on the greedy value function approximation (G-VFA), a KIP-specific approximation that relies on the fact that greedy algorithms are typically good for 1-dimensional knapsack problems. Namely, the heuristic is based on ordering the items with their value-to-weight ratio [15] and is used as the knapsack solution in the follower problem, while still being parameterized by $\mathbf{x}$. This heuristic is embedded in a single-level problem as this heuristic is MILP-representable [see 1]; we note that we are not aware of uses in the literature of this approximation and it may be of independent interest. Generally, G-VFA performs quite well, and in some cases outperforms $\mathrm{NN}^l$ and $\mathrm{NN}^u$, but there are clear cases where $\mathrm{NN}^l$ and $\mathrm{NN}^u$ outperform G-VFA demonstrating that learning is beneficial. In addition, heuristics like G-VFA can be utilized to compute features for $\mathrm{NN}^l$ and $\mathrm{NN}^u$. For KIP, the inclusion of these features derived from G-VFA strongly improves the results (see

Table 10 in Appendix I.3). This demonstrates that there is value in leveraging any problem-specific MILP-representable heuristics as features for learning.

**Q5: How does $\lambda$ affect $NN^l$?**  Table 9 in Appendix I.2 shows that a slack penalty of $\lambda = 0.1$ improves the performance of $NN^l$ on some instances for DNDP, compared to the $\lambda = 1$ reported in Table 2, indicating that tuning over $\lambda$ might be beneficial. As an alternative to adding slack, one can even dampen predictions of the value function to allow more flexibility using the empirical error observed during training; see Table 8 in Appendix I.1.

## 6   Related Work

**Learning for bilevel optimization.**  Besides the approaches of Sinha et al. [50, 51, 52] and Beykal et al. [9] discussed in Section 2, other learning-based methods have been introduced to solve BiLO problems. Bagloee et al. [2] present a heuristic for DNDP which uses a linear prediction of the leader's objective function. An iterative algorithm refines the prediction with new solutions, terminating after a pre-determined number of iterations. Chan et al. [13] propose to simultaneously optimize the parameters of a learning model for a subset of followers in a large-scale cycling network design problem. Here, only non-parametric or linear models are utilized as optimizing more sophisticated learning models is generally challenging with MILP-based optimization. Molan and Schmidt [42] make use of a neural network to predict the follower variables. The authors assume a setting with a black-box follower's problem, no coupling constraints, and continuous leader variables. Another learning-based heuristic is proposed by Kwon et al. [35] for a bilevel knapsack problem. This approach is knapsack-specific and requires a sophisticated, GPU-based, problem-specific graph neural network for which no code is publicly available. Zhou et al. [59] propose a learning-based algorithm for binary bilevel problems which, similar to our approach, predicts the optimal value function and develops a single-level reformulation based on the trained model. They propose using a graph neural network and an input-supermodular neural network, both of which can only be trained on a single instance rather than learning across classes of instances as NEUR2BILO does. NEUR2BILO significantly outperforms this method as shown in Appendix F. For continuous unconstrained bilevel optimization, a substantially different setting, many methods have been proposed recently due to interest in solving nested problems in machine learning (e.g., hyperparameter tuning and meta-learning) [37].

**Data-driven optimization.**  The integration of a trained machine learning model into a MIP is a vital element of NEUR2BILO. This is possible due to MILP formulations of neural networks [14, 22, 49], and of other predictors like decision trees [38, 8]. These methods have become easily applicable due to open software implementations [7, 12, 40, 55] and the `gurobi-machinelearning` library. One such application is constraint learning [21]. More similar to our setting are the approaches in [19, 20, 34] for predicting value functions of other nested problems such as two-stage stochastic and robust optimization. Our method caters to the specificities of BiLO, particularly in the lower-level approximation which performs well in highly-constrained BiLO settings such as the DNDP, has approximation guarantees based on the error of the predictive model, and computational results on problems with non-linear interactions between the variables in each stage of the optimization problem; these aspects distinguish NEUR2BILO from prior work.

## 7   Conclusion

In both its upper- and lower-level instantiations, NEUR2BILO finds high-quality solutions in a few milliseconds or seconds across four benchmarks that span applications in interdiction, network security, healthcare, and transportation planning. In fact, we are not aware of any bilevel optimization method which has been evaluated across such a diverse range of problems as existing methods make stricter assumptions that limit their applicability. NEUR2BILO models are generic, easy to train, and accommodating of problem-specific heuristics as features. One limitation of our experiments is that they lack a problem that involves coupling constraints in (1b). We could not identify benchmark problems with this property in the literature, but exploring this setting would be valuable. Of future interest are potential extensions to bilevel *stochastic* optimization [6], robust optimization with decision-dependent uncertainty [28] (a special case of BiLO), and multi-level problems beyond two levels, e.g. [36].

## Acknowledgments

Dumouchelle, Julien, and Khalil acknowledge funding support from the Natural Sciences and Engineering Research Council Discovery Grant Program and the SCALE AI Research Chair Program. Julien received funding from the Dutch Research Council (Nederlandse Organisatie voor Wetenschappelijk Onderzoek, NWO) under project OCENW.GROOT.2019.015.

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

## A  Impact Statement

Bilevel optimization has been used to model attacker-defender situations, which could be used in defense or similar political contexts. We have attempted to cover more socially beneficial healthcare and transportation planning applications but duly acknowledge that our methods could be applied in rather nefarious domains. That being said, there remain many domains that could benefit from our work and that are widely beneficial, such as in the management of energy systems.

## B  NEUR2BILO Pseudocode

Here, we outline pseudocode for NEUR2BILO. Algorithm 1 presents the pseudocode for data collection and training. Algorithm 2 presents the pseudocode for optimization. Following Algorithm 2, the objective is computed via the bilevel feasibility procedure detailed in Section 3.1. Note that data collection can be done once to collect labels for both the upper- and lower-level approximations. Additionally, a single trained model may be (and is in our experiments) evaluated across multiple test instances.

---
**Algorithm 1** NEUR2BILO Data Collection and Training

---

**Data Collection**
  $\mathcal{D} \leftarrow \{\}$
  **for** $i = 1$ **to** number of instances to sample **do**
    $\mathcal{P} \leftarrow$ sampled instance. Note that $\mathcal{P}$ is defined by $F(\cdot), G(\cdot), f(\cdot), g(\cdot), \mathcal{Y}$, and $\mathcal{X}$. For most BiLO problems, these functions are defined by the constraint and objective coefficients
    **for** $j = 1$ **to** number of decisions per-instance **do**
      $\mathbf{x} \leftarrow$ sampled upper-level decision
      $\mathbf{y}^{\star} \leftarrow \arg\max_{\mathbf{y} \in \mathcal{Y}} \{f(\mathbf{x}, \mathbf{y}) : g(\mathbf{x}, \mathbf{y}) \geq \mathbf{0}\}$
      Add $(\mathcal{P}, \mathbf{x}, F(\mathbf{x}, \mathbf{y}^{\star}), f(\mathbf{x}, \mathbf{y}^{\star}))$ to $\mathcal{D}$
    **end for**
  **end for**
  **return** $\mathcal{D}$

**Training**
  **if** approximating upper-level **then**
    Train regressor ($\text{NN}^{u}$) with features $(\mathcal{P}, \mathbf{x})$ and label $(F(\cdot))$ from dataset $\mathcal{D}$
  **else if** approximating lower-level **then**
    Train regressor ($\text{NN}^{l}$) with the features $(\mathcal{P}, \mathbf{x})$ and label $(f(\cdot))$ from dataset $\mathcal{D}$
  **end if**
  **return** $\text{NN}^{u}$ or $\text{NN}^{l}$

---

---
**Algorithm 2** NEUR2BILO Optimization

---

**Input:** Evaluation instance $\mathcal{P}'$, trained model $\text{NN}^{u}/\text{NN}^{u}$. Note that the trained model ($\text{NN}^{u}/\text{NN}^{u}$) is used on multiple evaluation instances.
**if** approximating upper-level **then**
  $\mathbf{x}^{\star} \leftarrow$ upper-level solution from the upper-level approximation (Equation (5))
**else if** approximating lower-level **then**
  $\mathbf{x}^{\star} \leftarrow$ upper-level solution from the lower-level approximation (Equation (7))
**end if**
**return** $\mathbf{x}^{\star}$

---

## C  Example Comparing the Upper- and Lower-level Approximations

**Example C.1.** Consider the problem

$$\min_{x \in \{0,1\}} y$$
$$s.t. \quad y \in \arg\max_{y \in \{0,1\}} \{y : 2x + y \le 1\}.$$

Solution $x = 1$ makes the follower's problem infeasible. For solution $x = 0$, the optimal follower solution is $y = 1$ leading to the optimal value 1. Assume that the same trained neural network is used in both approaches; this is possible since leader and follower have the same objective functions. If it predicts $\text{NN}(0) = 2$ and $\text{NN}(1) = 0$, then the upper-level approximation problem (5) will return $x = 1$ which is infeasible whereas the lower-level approximation (7) correctly returns $x = 0$.

## D  Proofs for Approximation Guarantees

This section includes the full analysis of the derived approximation guarantee in Section 3.3 for the lower-level approximation with $\text{NN}^l(\mathbf{x}; \Theta)$.

Recall that we look at a specific setup for which we derive approximation guarantees: the leader and the follower have the same objective function (i.e., $f(\mathbf{x}, \mathbf{y}) = F(\mathbf{x}, \mathbf{y})$ for all $\mathbf{x} \in \mathcal{X}, \mathbf{y} \in \mathcal{Y}$), we assume that Assumption 1(i) holds and that the neural network approximates the optimal value of the follower's problem up to an absolute error of $\alpha > 0$, i.e.,

$$|\text{NN}^l(\mathbf{x}; \Theta) - \Phi(\mathbf{x})| \le \alpha \quad \text{for all } \mathbf{x} \in \mathcal{X}. \tag{9}$$

We furthermore define the parameter $\Delta$ as the maximum difference of functions values $f(\mathbf{x}, \mathbf{y}) - f(\mathbf{x}, \mathbf{y}') \ge 0$ over all $\mathbf{x} \in \mathcal{X}, \mathbf{y}, \mathbf{y}' \in \mathcal{Y}$ such that no $\tilde{\mathbf{y}} \in \mathcal{Y}$ exists which has function value $f(\mathbf{x}, \mathbf{y}) > f(\mathbf{x}, \tilde{\mathbf{y}}) > f(\mathbf{x}, \mathbf{y}')$. Note that $\Delta$ can be strictly larger than zero if the follower decisions are integer.

For a fixed $\mathbf{x} \in \mathcal{X}$, $\mathbf{y}_{\text{NN}}^\star(\mathbf{x})$ denotes an optimal solution of (7). Furthermore, for any given $\mathbf{y} \in \mathcal{Y}$ we denote by $\mathbf{s}^\star(\mathbf{x}, \mathbf{y})$ an optimal slack-value in Problem (7) if the upper- and lower-level variables are fixed to $\mathbf{x}$ and $\mathbf{y}$, respectively.

**Observation D.1.** *For any $\mathbf{x} \in \mathcal{X}$ and $\mathbf{y} \in \mathcal{Y}$ we have*

$$\mathbf{s}^\star(\mathbf{x}, \mathbf{y}) = \max\{0, \text{NN}^l(\mathbf{x}; \Theta) - f(\mathbf{x}, \mathbf{y})\}.$$

**Lemma D.2.** *Assume the leader and the follower have the same objective function and $\lambda > 1$. Then, for any given $\mathbf{x} \in \mathcal{X}$ the following conditions hold for the optimal follower solution $\mathbf{y}_{\text{NN}}^\star(\mathbf{x})$ of Problem* (7)*:*

- *If $\text{NN}^l(\mathbf{x}; \Theta) \ge \Phi(\mathbf{x})$, then $f(\mathbf{x}, \mathbf{y}_{\text{NN}}^\star(\mathbf{x})) = \Phi(\mathbf{x})$, i.e., $(\mathbf{x}, \mathbf{y}_{\text{NN}}^\star(\mathbf{x}))$ is feasible for the original bilevel problem.*
- *If $\text{NN}^l(\mathbf{x}; \Theta) < \Phi(\mathbf{x})$, then $\text{NN}^l(\mathbf{x}; \Theta) - \frac{1}{\lambda}\Delta \le f(\mathbf{x}, \mathbf{y}_{\text{NN}}^\star(\mathbf{x})) \le \Phi(\mathbf{x})$.*

*Proof.* Case 1: Let $\mathbf{x} \in \mathcal{X}$ for which it holds $\text{NN}^l(\mathbf{x}; \Theta) \ge \Phi(\mathbf{x})$ and assume the opposite of the statement is true, i.e., for the optimal reaction $y_{\text{NN}}^\star(\mathbf{x})$ in (7) it holds that $\Phi(\mathbf{x}) > f(\mathbf{x}, \mathbf{y}_{\text{NN}}^\star(\mathbf{x}))$. Since $\lambda > 0$ and due to Constraint (7d) the optimal slack value for solution $\mathbf{x}$ in Problem (7) is $\mathbf{s}^\star(\mathbf{x}, \mathbf{y}) = \text{NN}^l(\mathbf{x}; \Theta) - f(\mathbf{x}, \mathbf{y})$. Assume $\mathbf{y}^\star(\mathbf{x})$ is the optimal follower reaction in (2) for $\mathbf{x}$, then it holds that:

$$f(\mathbf{x}, \mathbf{y}_{\text{NN}}^\star(\mathbf{x})) + \lambda \mathbf{s}^\star(\mathbf{x}, \mathbf{y}_{\text{NN}}^\star(\mathbf{x}))$$
$$= f(\mathbf{x}, \mathbf{y}_{\text{NN}}^\star(\mathbf{x})) + \lambda \left(\text{NN}^l(\mathbf{x}; \Theta) - f(\mathbf{x}, \mathbf{y}_{\text{NN}}^\star(\mathbf{x}))\right)$$
$$> f(\mathbf{x}, \mathbf{y}_{\text{NN}}^\star(\mathbf{x})) + \lambda \left(\text{NN}^l(\mathbf{x}; \Theta) - f(\mathbf{x}, \mathbf{y}_{\text{NN}}^\star(\mathbf{x}))\right) + (\lambda - 1)\left(f(\mathbf{x}, \mathbf{y}_{\text{NN}}^\star(\mathbf{x})) - f(\mathbf{x}, \mathbf{y}^\star(\mathbf{x}))\right)$$
$$= f(\mathbf{x}, \mathbf{y}^\star(\mathbf{x})) + \lambda \left(\text{NN}^l(\mathbf{x}; \Theta) - f(\mathbf{x}, \mathbf{y}^\star(\mathbf{x}))\right)$$
$$= f(\mathbf{x}, \mathbf{y}^\star(\mathbf{x})) + \lambda \mathbf{s}^\star(\mathbf{x}, \mathbf{y}^\star(\mathbf{x}))$$

where the first inequality follows since $\lambda > 1$ and $f(\mathbf{x}, \mathbf{y}^\star(\mathbf{x})) = \Phi(\mathbf{x}) > f(\mathbf{x}, \mathbf{y}^\star_{\text{NN}}(\mathbf{x}))$ and the latter equality follows from $\text{NN}^l(\mathbf{x}; \Theta) \geq \Phi(\mathbf{x}) = f(\mathbf{x}, \mathbf{y}^\star(\mathbf{x}))$. The latter result shows that the solution $(\mathbf{x}, \mathbf{y}^\star(\mathbf{x}))$ has a strictly better objective value in the surrogate problem (7) than $(\mathbf{x}, \mathbf{y}^\star_{\text{NN}}(\mathbf{x}))$ which contradicts the optimality of $(\mathbf{x}, \mathbf{y}^\star_{\text{NN}}(\mathbf{x}))$.

Case 2: Let $\mathbf{x} \in \mathcal{X}$ be a leader's decision for which $\text{NN}^l(\mathbf{x}; \Theta) < \Phi(\mathbf{x})$ and assume the opposite of the statement, i.e., for the optimal reaction $\mathbf{y}^\star_{\text{NN}}(\mathbf{x})$ in (7) it holds that $\text{NN}^l(\mathbf{x}; \Theta) - \frac{1}{\lambda}\Delta > f(\mathbf{x}, \mathbf{y}^\star_{\text{NN}}(\mathbf{x}))$. Hence the optimal slack value in (7) is

$$s^\star(\mathbf{x}, \mathbf{y}^\star_{\text{NN}}(\mathbf{x})) = \text{NN}^l(\mathbf{x}; \Theta) - f(\mathbf{x}, \mathbf{y}^\star_{\text{NN}}(\mathbf{x})) > \frac{1}{\lambda}\Delta. \tag{10}$$

First, assume there exists another feasible solution $\bar{\mathbf{y}}(\mathbf{x})$ for Problem (7) with

$$f(\mathbf{x}, \mathbf{y}^\star_{\text{NN}}(\mathbf{x})) < f(\mathbf{x}, \bar{\mathbf{y}}(\mathbf{x})) < \text{NN}^l(\mathbf{x}; \Theta)$$

then solution $(\mathbf{x}, \bar{\mathbf{y}}(x))$ has a strictly better objective value than $(\mathbf{x}, \mathbf{y}^\star_{\text{NN}}(\mathbf{x}))$ in (7) since increasing the value of $f$ by $\delta$ decreases the value of the slack variable by $\delta$ which results in a better objective value since $\lambda > 1$, which contradicts the optimality of $(\mathbf{x}, \mathbf{y}^\star_{\text{NN}}(\mathbf{x}))$.

Second, assume there exists no other feasible solution $\bar{\mathbf{y}}(\mathbf{x})$ for Problem (7) with

$$f(\mathbf{x}, \mathbf{y}^\star_{\text{NN}}(\mathbf{x})) < f(\mathbf{x}, \bar{\mathbf{y}}(\mathbf{x})) < \text{NN}^l(\mathbf{x}; \Theta).$$

Then there must exists a feasible solution $\bar{\mathbf{y}}(\mathbf{x})$ with $f(\mathbf{x}, \bar{\mathbf{y}}(\mathbf{x})) \geq \text{NN}^l(\mathbf{x}; \Theta)$ and

$$f(\mathbf{x}, \bar{\mathbf{y}}(\mathbf{x})) - f(\mathbf{x}, \mathbf{y}^\star_{\text{NN}}(\mathbf{x})) \leq \Delta, \tag{11}$$

by definition of $\Delta$. In this case, we have

$$\begin{aligned}
f(\mathbf{x}, \mathbf{y}^\star_{\text{NN}}(\mathbf{x})) &+ \lambda\mathbf{s}^\star(\mathbf{x}, \mathbf{y}^\star_{\text{NN}}(\mathbf{x})) - f(\mathbf{x}, \bar{\mathbf{y}}(\mathbf{x})) - \lambda\mathbf{s}^\star(\mathbf{x}, \bar{\mathbf{y}}(\mathbf{x})) \\
&= f(\mathbf{x}, \mathbf{y}^\star_{\text{NN}}(\mathbf{x})) + \lambda\mathbf{s}^\star(\mathbf{x}, \mathbf{y}^\star_{\text{NN}}(\mathbf{x})) - f(\mathbf{x}, \bar{\mathbf{y}}(\mathbf{x})) \\
&> f(\mathbf{x}, \mathbf{y}^\star_{\text{NN}}(\mathbf{x})) + \Delta - f(\mathbf{x}, \bar{\mathbf{y}}(\mathbf{x})) \geq -\Delta + \Delta = 0,
\end{aligned}$$

where the first equality follows since $\mathbf{s}^\star(\mathbf{x}, \bar{\mathbf{y}}(\mathbf{x})) = 0$, the first inequality follows from (10) and the last inequality follows from (11). In summary, the latter results show that there exists a solution $(\mathbf{x}, \bar{\mathbf{y}}(\mathbf{x}))$ for (7) which has strictly better objective value than $(\mathbf{x}, \mathbf{y}^\star_{\text{NN}}(\mathbf{x}))$ which is a contradiction.

Note that the inequality $f(\mathbf{x}, \mathbf{y}^\star_{\text{NN}}(\mathbf{x})) \leq \Phi(\mathbf{x})$ follows directly from the definiton of $\Phi(\mathbf{x})$. □

The latter lemma states, that if the neural network is overestimating the follower value for a solution $\mathbf{x} \in \mathcal{X}$, then the surrogate problem (7) still selects an optimal follower response. However, if the neural network underestimates the value, it may happen that the surrogate problem chooses a follower response for which the objective value either is larger than the true value or differs by at most $\frac{1}{\lambda}\Delta$. Note that the latter term can be controlled by increasing the penalty $\lambda$.

By applying Lemma D.2 we can bound the approximation guarantee of the lower-level NEUR2BILO.

**Theorem 3.1.** If the leader and the follower have the same objective function and $\lambda > 1$, NEUR2BILO returns a feasible solution $(\mathbf{x}^\star, \mathbf{y}^\star)$ for Problem (1) with objective value

$$f(\mathbf{x}^\star, \mathbf{y}^\star) \leq \text{opt} + 3\alpha + \frac{2}{\lambda}\Delta,$$

where opt is the optimal value of (1) and $\lambda$ the penalty term in (7a) .

*Proof.* Let $(\mathbf{x}^\star_{\text{NN}}, \mathbf{y}^\star_{\text{NN}})$ be an optimal solution of the surrogate problem (7). By Lemma D.2 and by definition (8) it follows that

$$\begin{aligned}
\Phi(\mathbf{x}^\star_{\text{NN}}) \geq f(\mathbf{x}^\star_{\text{NN}}, \mathbf{y}^\star_{\text{NN}}) &\geq \text{NN}^l(\mathbf{x}^\star_{\text{NN}}; \Theta) - \frac{1}{\lambda}\Delta \\
&\geq \Phi(\mathbf{x}^\star_{\text{NN}}) - \alpha - \frac{1}{\lambda}\Delta.
\end{aligned} \tag{12}$$

Following the three steps presented in Section 3.1, NEUR2BILO returns a feasible solution $(\mathbf{x}^\star, \mathbf{y}^\star)$ for Problem (2) where $\mathbf{x}^\star = \mathbf{x}^\star_{\text{NN}}$ and $f(\mathbf{x}^\star, \mathbf{y}^\star) = \Phi(\mathbf{x}^\star)$. Hence the following holds:

$$f(\mathbf{x}^\star, \mathbf{y}^\star) = \Phi(\mathbf{x}^\star) \leq f(\mathbf{x}^\star, \mathbf{y}^\star_{\text{NN}}) + \alpha + \frac{1}{\lambda}\Delta. \tag{13}$$

Assume $(\mathbf{x}^{\star\star}, \mathbf{y}^{\star\star})$ is an optimal bilevel solution of Problem (1) and $\mathbf{y}_{NN}^{\star\star}$ the optimal follower response in the surrogate problem (7). Then we have

$$f(\mathbf{x}^\star, \mathbf{y}_{NN}^\star) + \mathbf{s}^*(\mathbf{x}^\star, \mathbf{y}_{NN}^\star) \leq f(\mathbf{x}^{\star\star}, \mathbf{y}_{NN}^{\star\star}) + \mathbf{s}^*(\mathbf{x}^{\star\star}, \mathbf{y}_{NN}^{\star\star})$$

since $(\mathbf{x}_{NN}^\star, \mathbf{y}_{NN}^\star)$ is an optimal solution of (7) with objective value given by (7a). From the latter inequality we obtain

$$\begin{aligned}
f(\mathbf{x}^\star, \mathbf{y}_{NN}^\star) &\leq f(\mathbf{x}^{\star\star}, y_{NN}^{\star\star}) + \mathbf{s}^*(\mathbf{x}^{\star\star}, \mathbf{y}_{NN}^{\star\star}) - \mathbf{s}^*(\mathbf{x}^\star, \mathbf{y}_{NN}^\star) \\
&\leq f(\mathbf{x}^{\star\star}, \mathbf{y}^{\star\star}) + \mathbf{s}^*(\mathbf{x}^{\star\star}, \mathbf{y}_{NN}^{\star\star}) \\
&\leq f(\mathbf{x}^{\star\star}, \mathbf{y}^{\star\star}) + NN^l(\mathbf{x}^{\star\star}; \Theta) - f(\mathbf{x}^{\star\star}, \mathbf{y}_{NN}^{\star\star}) \\
&\leq f(\mathbf{x}^{\star\star}, \mathbf{y}^{\star\star}) + \Phi(\mathbf{x}^{\star\star}) + \alpha - (\Phi(\mathbf{x}^{\star\star}) - \alpha - \frac{1}{\lambda}\Delta) \\
&= \text{opt} + 2\alpha + \frac{1}{\lambda}\Delta
\end{aligned}$$

where the second inequality follows from $\mathbf{s}^*(\mathbf{x}^\star, \mathbf{y}_{NN}^\star) \geq 0$ and $\mathbf{y}^{\star\star}$ being an optimal follower solution for $\mathbf{x}^{\star\star}$. The third inequality follows from Observation D.1 and the fourth inequality follows from (8) and from (12) applied to $\mathbf{x}^{\star\star}$.

Together with (13), this completes the proof. □

# E    Problem Formulations

## E.1    Knapsack interdiction

The bilevel knapsack problem with interdiction constraints as described in Tang et al. [54] is given by

$$\begin{aligned}
\min_{\mathbf{x} \in \{0,1\}^n, \mathbf{y}} \quad & \sum_{i=1}^n p_i y_i \\
\text{s.t.} \quad & \sum_{i=1}^n x_i \leq k, \\
& \mathbf{y} \in \arg\max_{\mathbf{y}' \in \{0,1\}^n} \quad \sum_{i=1}^n p_i y_i' \\
& \qquad\qquad \text{s.t.} \quad \sum_{i=1}^n a_i y_i' \leq b, \\
& \qquad\qquad\qquad\quad y_i' + x_i \leq 1, i \in [n],
\end{aligned}$$

where $\mathbf{x}$ are the leader's variables and $\mathbf{y}$ are that of the follower. The leader decides to interdict (a maximum of $k$) items of the knapsack solved in the follower's problem with $n$ the number of items, $p_i$ the profits, $a_i$ the weight of item $i$, respectively, and the budget of the knapsack is denoted by $b$.

## E.2    Critical node problem

The critical node problem is described in Carvalho et al. [11] as follows

$$\begin{aligned}
\max_{\mathbf{x} \in \{0,1\}^n, \mathbf{y}} \quad & \sum_{i=1}^n \left( p_i^d \big((1-x_i)(1-y_i) + \eta x_i y_i + \epsilon x_i(1-y_i) + \delta(1-x_i)y_i\big) \right) \\
\text{s.t.} \quad & \sum_{i=1}^n d_i x_i \leq D, \\
& \mathbf{y} \in \arg\max_{\mathbf{y}' \in \{0,1\}^n} \quad \sum_{i=1}^n \left( p_i^a \big(-\gamma(1-x_i)(1-y_i') + (1-x_i)y_i' + (1-\eta)x_i y_i'\big) \right) \\
& \qquad\qquad \text{s.t.} \quad \sum_{i=1}^n a_i y_i' \leq A,
\end{aligned}$$

where $\mathbf{x}$ and $\mathbf{y}$ are the leader's and follower's variables, respectively. Here, $\mathbf{x}$ denotes the decisions of the leader (defender) who selects which nodes to deploy resources to defend a set of nodes, while $\mathbf{y}$ are the decisions for the follower (attacker) for which nodes to attack. $d_i$ and $a_i$ are the costs for the $x_i$ and $y_i$, respectively. $D$ and $A$ are the budgets for the defender and attacker, respectively. In this problem, the bilinearity arises in the objectives of both the leader and follower, which results in four outcomes for each possible combination of defending and attacking a node $i$. The first outcome arises when both the leader and follower do not select the node. In this case, the leader receives the full profit, $p_i^d$, and the follower pays an opportunity cost of $-\gamma p_i^a$ for not attacking an undefended node. Second is a successful attack, wherein the leader receives a reduced profit of $\delta p_i^d$ and the follower receives the full profit $p_i^a$. Third is a mitigated attack, wherein the leader receives a profit of $\eta p_i^d$ for a degradation in operations, while the follower receives a profit of $(1 - \eta)p_i^a$ for a mitigated attack. Fourth is a mitigation without an attack, wherein the leader receives a profit $\epsilon p_i^d$ for a degradation in operations, while the follower receives a profit of 0 for a mitigated attack.

### E.3 Donor-recipient problem

The donor-recipient problem as described in Ghatkar et al. [27], and introduced in Morton et al. [43], is formulated as

$$
\begin{aligned}
\max_{\mathbf{x} \in [0,1]^n, \mathbf{y}, y_0} \quad & \sum_{i=1}^{n} w_i y_i \\
\text{s.t.} \quad & \sum_{i=1}^{n} c_i x_i \leq B_d, \\
& (\mathbf{y}, y_0) \in \underset{\mathbf{y}' \in \{0,1\}^n, y_0' \in [0,1]}{\arg\max} \quad \sum_{i=1}^{n} v_i y_i' + v_0 y_0' \\
& \qquad\qquad \text{s.t.} \quad \sum_{i=1}^{n} (c_i - c_i x_i) y_i' + c_0 y_0' \leq B_r,
\end{aligned}
$$

where the leader's decisions $\mathbf{x}$ represent those of the donor and the follower's decisions $(\mathbf{y}, y_0)$ the ones of the recipient. The profit of project $i$ is given as $w_i$ for the leader and $v_i$ for the follower, the cost as $c_i$, and the budget of the leader, resp. follower, as $B_d$ and $B_r$. Next to the projects, the recipient can allocate its budget to external projects, for which the profit is given as $v_0$ and the cost $c_0$.

### E.4 Discrete network design problem

We use the standard formulation from Section 1 following the computational benchmarking study of Rey [47] and the code provided by the author [2].

## F Comparison to the Learning-Based Approach of Zhou et al. [59]

This section compares our approach to a recent learning-based approach from Zhou et al. [59] based on code provided by the author [3]. We specifically compare the input-supermodular neural network (ISNN), i.e., the best-performing model from Zhou et al. [59]. Their approach requires sampling and training for each instance, which is reflected in the time, whereas the model for $\text{NN}^l$ and $\text{NN}^u$ can be trained once and evaluated across multiple instances, so the data collection and training time are excluded. We also restrict ISNN to run for one iteration given Zhou et al. [59] report very minimal improvements when increasing the number of iterations. Moreover, one iteration requires the least amount of time. Table 3 reports the MRE and time for each method for the knapsack instances from Tang et al. [54]. Generally, we can see a significant improvement over ISNN in both computing time and MRE.

---

[2] `https://github.com/davidrey123/DNDP/`
[3] `https://github.com/bozlamberth/LearnBilevel/`

| $n$ | $k$ | ISNN | | $\mathrm{NN}^l$ | | $\mathrm{NN}^u$ | | G-VFA | | B&C | |
|---|---|---|---|---|---|---|---|---|---|---|---|
| | | MRE | Time | MRE | Time | MRE | Time | MRE | Time | MRE | Time |
| 18 | 5 | 10.50 | 254.35 | 1.48 | 0.59 | 1.48 | 0.34 | 1.82 | **0.14** | **0.00** | 9.55 |
| 18 | 9 | 46.50 | 227.49 | 1.51 | 0.59 | 1.51 | 0.43 | 3.97 | **0.22** | **0.00** | 5.81 |
| 18 | 14 | 302.10 | 217.62 | **0.00** | 0.22 | **0.00** | 0.17 | 64.22 | **0.03** | **0.00** | 0.39 |
| 20 | 5 | 8.56 | 262.01 | 0.41 | 0.62 | 0.41 | 0.45 | 2.19 | **0.25** | **0.00** | 23.18 |
| 20 | 10 | 54.07 | 236.74 | 0.99 | 0.66 | 0.99 | 0.58 | 0.99 | **0.36** | **0.00** | 10.27 |
| 20 | 15 | 447.29 | 229.41 | 3.57 | 0.32 | 3.57 | 0.19 | 23.39 | **0.02** | **0.00** | 0.94 |
| 22 | 6 | 17.32 | 266.55 | 0.71 | 0.19 | 0.71 | **0.18** | 0.42 | 0.18 | **0.00** | 42.30 |
| 22 | 11 | 66.24 | 247.97 | 1.01 | 0.28 | 1.01 | **0.28** | 1.08 | 0.33 | **0.00** | 16.26 |
| 22 | 17 | 485.75 | 241.61 | 14.43 | 0.24 | 14.43 | 0.15 | 14.43 | **0.13** | **0.00** | 0.68 |
| 25 | 7 | 14.75 | 280.71 | 0.44 | 2.66 | 0.44 | 2.42 | 3.85 | **0.64** | **0.00** | 137.96 |
| 25 | 13 | 61.57 | 264.09 | 1.42 | 2.75 | 1.42 | 2.79 | 3.85 | **1.24** | **0.00** | 48.43 |
| 25 | 19 | 424.92 | 262.04 | 2.49 | 0.48 | 2.49 | 0.38 | 2.49 | **0.13** | **0.00** | 1.77 |
| 28 | 7 | 19.17 | 297.44 | 0.39 | 0.67 | 0.39 | 0.74 | 0.26 | **0.62** | **0.00** | 309.18 |
| 28 | 14 | 73.61 | 286.02 | 0.75 | 2.10 | 0.75 | 1.45 | 1.37 | **1.29** | **0.00** | 120.74 |
| 28 | 21 | 423.15 | 279.51 | 1.14 | 0.45 | 1.14 | 0.49 | 3.16 | **0.31** | **0.00** | 4.92 |
| 30 | 8 | 21.01 | 305.67 | **0.00** | 1.54 | **0.00** | 1.54 | 0.43 | **0.97** | **0.00** | 792.44 |
| 30 | 15 | 68.19 | 295.92 | 0.49 | 3.64 | 0.49 | 3.06 | 0.75 | **1.35** | **0.00** | 187.23 |
| 30 | 23 | 416.03 | 290.94 | 2.29 | 1.08 | 2.29 | 0.73 | 4.48 | **0.25** | **0.00** | 5.65 |
| Average [54] | | 164.49 | 263.67 | 1.86 | 1.06 | 1.86 | 0.91 | 7.21 | **0.47** | **0.00** | 95.43 |

Table 3: Comparison to ISNN from Zhou et al. [59] on the knapsack interdiction problem. $n$ and $k$ denote the number of items and the interdiction budget, respectively. We directly evaluate on the 180 instances (10 per size) of Tang et al. [54]; each value is the average over 10 instances. We compare the upper- and lower-level approximations, as well as the no-learning baseline (G-VFA) and the exact algorithm (B&C).

# G   Objective & Incumbent Results

This section reports the more detailed information related to the objective values for each problem. Objective results for each problem are given in Tables 4-7. In addition, for KIP and CNP, as the solver from Fischetti et al. [23] provides easily accessible incumbent solutions, we include two additional metrics.

– The first metric "Solver Time Ratio" measures the time it takes the solver to obtain an equally good (or better) incumbent solution, divided by the solving time of the respective approximation. The number in brackets to the right indicates the number of instances for which the solver finds an equivalent solution.

– The second metric "Solver Relative Error at Time" measures the relative error of the best solution found by the solver compared to the respective approximation. The value in brackets to the right indicates the number of instances for which the solver finds an incumbent before the approximation is done solving.

| $n$ | $k$ | Objective | | | | Mean Relative Error (%) | | | | Solving Time | | | | Solver Time Ratio | | | Solver Relative Error at Time | | |
|---|---|---|---|---|---|---|---|---|---|---|---|---|---|---|---|---|---|---|---|
| | | $\mathrm{NN}^l$ | $\mathrm{NN}^u$ | G-VFA | B&C | $\mathrm{NN}^l$ | $\mathrm{NN}^u$ | G-VFA | B&C | $\mathrm{NN}^l$ | $\mathrm{NN}^u$ | G-VFA | B&C | $\mathrm{NN}^l$ | $\mathrm{NN}^u$ | G-VFA | $\mathrm{NN}^l$ | $\mathrm{NN}^u$ | G-VFA |
| 18 | 5 | 308.30 | 308.30 | 309.20 | **303.50** | 1.48 | 1.48 | 1.82 | **0.00** | 0.59 | 0.34 | **0.14** | 9.55 | 24.48 (10) | 35.86 (10) | 177.39 (10) | - (0) | - (0) | - (0) |
| 18 | 9 | 145.60 | 145.60 | 149.10 | **143.40** | 1.51 | 1.51 | 3.97 | **0.00** | 0.59 | 0.43 | **0.22** | 5.81 | 12.31 (10) | 18.52 (10) | 73.49 (10) | - (0) | - (0) | - (0) |
| 18 | 14 | **31.00** | **31.00** | 51.40 | **31.00** | **0.00** | **0.00** | 64.22 | **0.00** | 0.22 | 0.17 | **0.03** | 0.39 | 1.87 (10) | 2.86 (10) | 31.9 (10) | 44.0 (2) | 41.5 (2) | - (0) |
| 20 | 5 | 390.30 | 390.30 | 397.60 | **388.50** | 0.41 | 0.41 | 2.19 | **0.00** | 0.62 | 0.45 | **0.25** | 23.18 | 50.68 (10) | 65.56 (10) | 420.04 (10) | - (0) | - (0) | - (0) |
| 20 | 10 | 165.40 | 165.40 | 165.40 | **163.70** | 0.99 | 0.99 | 0.99 | **0.00** | 0.66 | 0.58 | **0.36** | 10.27 | 18.39 (10) | 22.03 (10) | 80.0 (10) | - (0) | - (0) | - (0) |
| 20 | 15 | 33.40 | 33.40 | 41.90 | **31.40** | 3.57 | 3.57 | 23.39 | **0.00** | 0.32 | 0.19 | **0.02** | 0.94 | 2.82 (10) | 4.75 (10) | 54.29 (10) | - (0) | - (0) | - (0) |
| 22 | 6 | 385.50 | 385.50 | 384.30 | **382.70** | 0.71 | 0.71 | 0.42 | **0.00** | 0.19 | 0.18 | 0.18 | 42.30 | 228.97 (10) | 249.8 (10) | 714.31 (10) | - (0) | - (0) | - (0) |
| 22 | 11 | 163.20 | 163.20 | 163.30 | **161.00** | 1.01 | 1.01 | 1.08 | **0.00** | 0.28 | 0.28 | 0.33 | 16.26 | 69.05 (10) | 74.99 (10) | 129.04 (10) | - (0) | - (0) | - (0) |
| 22 | 17 | 35.20 | 35.20 | 35.20 | **29.20** | 14.43 | 14.43 | 14.43 | **0.00** | 0.24 | 0.15 | **0.13** | 0.68 | 3.09 (10) | 5.48 (10) | 29.34 (10) | 18.0 (1) | - (0) | - (0) |
| 25 | 7 | 438.20 | 438.20 | 438.20 | **436.20** | 0.44 | 0.44 | 0.44 | **0.00** | 2.66 | 2.42 | **0.64** | 137.96 | 58.27 (10) | 61.24 (10) | 1102.38 (10) | 28.69 (2) | 28.69 (2) | 28.69 (2) |
| 25 | 13 | 194.90 | 194.90 | 199.90 | **191.50** | 1.42 | 1.42 | 3.85 | **0.00** | 2.75 | 2.79 | **1.24** | 48.43 | 21.14 (10) | 25.13 (10) | 67.86 (10) | - (0) | - (0) | - (0) |
| 25 | 19 | 43.30 | 43.30 | 43.30 | **41.80** | 2.49 | 2.49 | 2.49 | **0.00** | 0.48 | 0.38 | **0.13** | 1.77 | 3.98 (10) | 4.81 (10) | 38.29 (10) | - (0) | - (0) | - (0) |
| 28 | 7 | 518.30 | 518.30 | 517.60 | **516.10** | 0.39 | 0.39 | 0.26 | **0.00** | 0.67 | 0.74 | **0.62** | 309.18 | 671.57 (10) | 518.35 (10) | 1033.45 (10) | 29.28 (8) | 29.28 (8) | 29.48 (8) |
| 28 | 14 | 224.90 | 224.90 | 226.80 | **223.40** | 0.75 | 0.75 | 1.37 | **0.00** | 2.10 | 1.45 | **1.29** | 120.74 | 59.99 (10) | 84.36 (10) | 120.05 (10) | 19.84 (2) | 19.84 (2) | 16.14 (2) |
| 28 | 21 | 46.70 | 46.70 | 48.20 | **46.20** | 1.14 | 1.14 | 3.16 | **0.00** | 0.45 | 0.49 | **0.31** | 4.92 | 12.95 (10) | 11.66 (10) | 38.98 (10) | - (0) | - (0) | - (0) |
| 30 | 8 | **536.30** | **536.30** | 538.70 | **536.30** | **0.00** | **0.00** | 0.43 | **0.00** | 1.54 | 1.54 | **0.97** | 792.44 | 497.06 (10) | 455.29 (10) | 1924.47 (10) | 27.07 (10) | 27.07 (10) | 26.58 (10) |
| 30 | 15 | 231.20 | 231.20 | 231.90 | **230.00** | 0.49 | 0.49 | 0.75 | **0.00** | 3.64 | 3.06 | **1.35** | 187.23 | 56.14 (10) | 66.07 (10) | 254.88 (10) | 86.11 (6) | 86.11 (6) | 85.19 (6) |
| 30 | 23 | 49.00 | 49.00 | 50.40 | **47.50** | 2.29 | 2.29 | 4.48 | **0.00** | 1.08 | 0.73 | **0.25** | 5.65 | 7.5 (10) | 8.79 (10) | 48.27 (10) | - (0) | - (0) | - (0) |
| 100 | 25 | 2,164.71 | 2,164.69 | **2,145.07** | 2,318.99 | 0.93 | 0.93 | **0.00** | 8.09 | 10.02 | 8.40 | **4.19** | 3,600.40 | - (0) | - (0) | - (0) | 34.36 (100) | 34.99 (100) | 37.93 (100) |
| 100 | 50 | 965.37 | 965.37 | **956.76** | 1,043.71 | 0.96 | 0.96 | **0.04** | 8.96 | 51.68 | **49.28** | 53.74 | 3,600.44 | 23.56 (5) | 26.24 (5) | - (0) | 59.36 (100) | 60.81 (100) | 60.48 (100) |
| 100 | 75 | **245.01** | **245.01** | 245.08 | 259.95 | **0.08** | **0.08** | 0.12 | 5.87 | 24.69 | **23.78** | 35.27 | 3,600.52 | 133.35 (4) | 152.72 (4) | 138.07 (5) | 177.01 (100) | 196.86 (100) | 193.94 (100) |

Table 4: KIP objective and incumbent results. Each row averaged over 10 instances, except for $n = 100$, which is average over 100 instances. $\mathrm{NN}^l$ and $\mathrm{NN}^u$ specify the lower- and upper-level approximations respectively. All times in seconds.

| $|V|$ | Objective | | | Mean Relative Error (%) | | | Times | | | Solver Time Ratio | | Solver Relative Error at Time | |
|---|---|---|---|---|---|---|---|---|---|---|---|---|---|
| | $NN^l$ | $NN^u$ | B&C | $NN^l$ | $NN^u$ | B&C | $NN^l$ | $NN^u$ | B&C | $NN^l$ | $NN^u$ | $NN^l$ | $NN^u$ |
| 10 | 224.47 | 225.10 | **228.63** | 3.20 | 2.75 | **1.01** | 1.69 | **1.19** | 3,600.80 | 136.66 (288) | 191.34 (289) | 269.0 (1) | - (0) |
| 25 | 562.72 | 566.23 | **572.51** | 2.60 | 1.77 | **0.73** | 1.69 | **1.19** | 3,600.80 | 736.84 (275) | 3934.02 (271) | 2.22 (248) | 2.57 (124) |
| 50 | 1,139.27 | 1,143.95 | **1,148.17** | 1.42 | 0.98 | **0.67** | 1.69 | **1.19** | 3,600.80 | 718.74 (225) | 3840.41 (183) | 1.94 (295) | 3.17 (190) |
| 100 | 2,285.15 | **2,297.47** | 2,272.30 | 1.12 | **0.56** | 1.79 | 1.69 | **1.19** | 3,600.80 | 645.37 (131) | 926.6 (90) | 2.4 (283) | 2.96 (278) |
| 300 | 6,781.91 | **6,882.42** | 6,755.07 | 2.01 | **0.33** | 2.32 | 1.69 | **1.19** | 3,600.80 | 41.65 (166) | 167.38 (47) | 1.49 (245) | 2.65 (243) |
| 500 | 11,348.60 | **11,439.25** | 11,208.43 | 1.33 | **0.45** | 2.47 | 1.69 | **1.19** | 3,600.80 | 106.9 (83) | 99.48 (15) | 1.51 (206) | 2.45 (205) |

Table 5: CNP objective and incumbent results. Each row averaged over 300 instances. All times in seconds.

| Instance # | Objective | | | Relative Error (%) | | | Times | | |
|---|---|---|---|---|---|---|---|---|---|
| | $NN^l$ | $NN^u$ | B&C+ | $NN^l$ | $NN^u$ | B&C+ | $NN^l$ | $NN^u$ | B&C+ |
| 1 | 34,356.00 | **59,524.00** | 47,206.00 | 42.28 | **0.00** | 20.69 | **0.09** | 1.44 | 3,600.09 |
| 2 | 33,713.00 | **54,764.00** | 39,526.00 | 38.44 | **0.00** | 27.82 | **0.12** | 1.52 | 3,600.08 |
| 3 | 36,717.00 | **66,967.00** | 46,792.00 | 45.17 | **0.00** | 30.13 | **0.14** | 2.85 | 3,600.07 |
| 4 | 36,414.00 | **54,908.00** | 44,486.00 | 33.68 | **0.00** | 18.98 | **0.07** | 1.68 | 3,637.23 |
| 5 | 33,090.00 | **59,627.00** | 43,355.00 | 44.51 | **0.00** | 27.29 | **0.10** | 1.96 | 3,600.07 |
| 6 | 36,691.00 | **56,603.00** | 39,006.00 | 35.18 | **0.00** | 31.09 | **0.08** | 2.93 | 3,600.10 |
| 7 | 31,354.00 | **55,569.00** | 43,443.00 | 43.58 | **0.00** | 21.82 | **0.09** | 1.58 | 3,600.14 |
| 8 | 35,710.00 | **54,414.00** | 39,839.00 | 34.37 | **0.00** | 26.79 | **0.09** | 0.87 | 3,600.10 |
| 9 | 38,961.00 | **61,869.00** | 45,288.00 | 37.03 | **0.00** | 26.80 | **0.16** | 4.55 | 3,600.16 |
| 10 | 36,965.00 | **60,488.00** | 43,194.00 | 38.89 | **0.00** | 28.59 | **0.12** | 3.57 | 3,600.10 |
| Averaged | 35,397.10 | **58,473.30** | 43,213.50 | 39.31 | **0.00** | 26.00 | **0.11** | 2.30 | 3,603.82 |

Table 6: DRP objective results. Each row corresponds to a single instance from dataset 15, i.e., the most challenging instances from Ghatkar et al. [27]. All times in seconds.

| # of edges | budget | Objective | | | | | | | Relative Error (%) | | | | | | | Times | | | |
|---|---|---|---|---|---|---|---|---|---|---|---|---|---|---|---|---|---|---|---|
| | | $NN^l$ | $NN^u$ | $GBT^l$ | $GBT^u$ | MKKT-5 | MKKT-10 | MKKT-30 | $NN^l$ | $NN^u$ | $GBT^l$ | $GBT^u$ | MKKT-5 | MKKT-10 | MKKT-30 | $NN^l$ | $NN^u$ | $GBT^l$ | $GBT^u$ |
| 10 | 0.25 | 6,201.25 | 6,145.27 | 6,214.37 | 6,147.02 | 6,484.98 | 6,155.69 | **6,129.65** | 1.21 | 0.36 | 1.43 | 0.38 | 5.78 | 0.51 | **0.10** | 2.95 | **0.01** | 3.19 | 0.09 |
| 10 | 0.5 | 5,532.92 | 5,557.28 | 5,531.27 | 5,640.77 | 5,849.03 | 5,618.41 | **5,492.23** | 0.73 | 1.22 | 0.72 | 2.74 | 6.47 | 2.17 | **0.00** | 3.35 | **0.01** | 3.66 | 0.07 |
| 10 | 0.75 | 5,202.82 | 5,246.29 | 5,211.07 | 5,225.02 | 5,477.72 | **5,179.39** | 5,181.30 | 0.47 | 1.32 | 0.63 | 0.91 | 5.80 | **0.02** | 0.06 | 2.80 | **0.00** | 3.02 | 0.06 |
| 20 | 0.25 | 5,478.52 | 5,272.98 | 5,272.07 | 5,210.07 | 5,535.14 | 5,423.02 | **5,180.67** | 6.05 | 2.64 | 2.38 | 1.41 | 7.78 | 5.12 | **0.85** | 5.02 | **0.02** | 5.02 | 0.23 |
| 20 | 0.5 | 4,347.58 | 4,490.04 | 4,356.47 | 4,390.52 | 4,563.35 | 4,416.83 | **4,330.21** | 1.01 | 4.36 | 1.22 | 2.02 | 6.00 | 2.52 | **0.64** | 4.91 | **0.03** | 5.02 | 0.21 |
| 20 | 0.75 | 4,084.19 | 4,085.09 | 4,061.68 | 4,135.70 | 4,363.00 | 4,057.72 | **4,053.02** | 0.85 | 0.91 | 0.32 | 2.14 | 7.87 | 0.22 | **0.11** | 4.47 | **0.01** | 4.69 | 0.13 |

Table 7: DNDP objective results. Each is averaged across 10 instances. All times in seconds.

## H   Distributional Results for Relative Error

## I   Ablation

### I.1   Lower-level value function constraints

In this section, we present an ablation study comparing alternative types of value function approximation (VFA) for the lower-level approximation on the KIP. Namely, we compare the approach used the the main paper, $NN^l$, which utilizes a slack variable to ensure feasibility. In addition, we include $NN^n$ which does not use a slack at all, and $NN^d$, which uses the largest error in the validation set to scale the prediction down. Table 8 reports objectives, relative errors, and solving times of each method. In general, the solution quality of $NN^l$ slightly exceeds that of $NN^d$, while $NN^n$ does significantly worse. The latter results is unsurprising given that any underestimation will cause a loss of feasibility for potentially high quality upper-level decisions. $NN^l$ is additionally generally the fastest to optimize as well.

### I.2   The effect of $\lambda$

In this section, we present results with $\lambda = 0.1$ for DNDP. Table 9 presents relative error and solving times for this setting. Notably, this choice of $\lambda$ tends to provide higher quality solutions than $\lambda = 1$, as reported in the main paper in Table 2. Tuning this hyperparameter further can thus improve the already strong numerical results reported for DNDP, and possibly other problems.

### I.3   Greedy features for Knapsack

This section explores the impact of the use of greedy features on the KIP problem. We specifically compare a model trained purely on the coefficients to a model trained on the coefficients with additional features derived from KIP-specific greedy heuristics. From Table 10, there is a clear advantage with the greedy features in terms of solution quality at the cost of increased solving time.

| $n$ | $k$ | Objective | | | Mean Relative Error (%) | | | Times | | |
|---|---|---|---|---|---|---|---|---|---|---|
| | | $NN^l$ | $NN^d$ | $NN^n$ | $NN^l$ | $NN^d$ | $NN^n$ | $NN^l$ | $NN^d$ | $NN^n$ |
| 18 | 5 | **308.30** | 308.40 | 318.40 | **0.00** | 0.03 | 3.28 | **0.59** | 0.83 | 1.06 |
| 18 | 9 | **145.60** | 145.60 | 152.90 | **0.00** | 0.00 | 6.70 | **0.59** | 1.21 | 0.81 |
| 18 | 14 | **31.00** | 37.50 | 40.00 | **0.00** | 16.91 | 48.23 | **0.22** | 0.32 | 0.35 |
| 20 | 5 | **390.30** | 390.30 | 413.90 | **0.00** | 0.00 | 6.48 | **0.62** | 0.79 | 1.38 |
| 20 | 10 | **165.40** | 165.40 | 175.90 | **0.00** | 0.00 | 6.60 | **0.66** | 1.47 | 1.76 |
| 20 | 15 | 33.40 | **32.50** | 55.70 | 3.33 | 14.29 | 100.91 | **0.32** | 0.38 | 0.96 |
| 22 | 6 | **385.50** | 386.80 | 403.00 | **0.00** | 0.27 | 4.56 | **0.19** | 0.37 | 0.80 |
| 22 | 11 | 163.20 | **162.10** | 179.20 | 0.55 | **0.07** | 11.83 | **0.28** | 0.85 | 1.23 |
| 22 | 17 | **35.20** | 35.20 | 49.00 | 5.15 | **4.63** | 69.91 | 0.24 | **0.19** | 0.41 |
| 25 | 7 | **438.20** | 438.20 | 446.50 | **0.00** | 0.00 | 1.98 | 2.66 | **2.40** | 3.85 |
| 25 | 13 | **194.90** | 195.50 | 206.50 | **0.00** | 0.26 | 6.67 | **2.75** | 3.25 | 4.83 |
| 25 | 19 | **43.30** | 43.30 | 64.40 | 1.69 | 1.69 | 92.49 | **0.48** | 0.74 | 1.84 |
| 28 | 7 | **518.30** | 518.30 | 532.20 | **0.00** | 0.00 | 2.80 | **0.67** | 0.83 | 2.37 |
| 28 | 14 | **224.90** | 224.90 | 234.70 | **0.00** | 0.00 | 4.60 | **2.10** | 2.69 | 3.72 |
| 28 | 21 | **46.70** | 49.90 | 60.70 | **0.00** | 7.48 | 37.45 | **0.45** | 0.83 | 1.67 |
| 30 | 8 | **536.30** | 537.10 | 537.70 | **0.00** | 0.18 | 0.25 | **1.54** | 1.86 | 3.07 |
| 30 | 15 | **231.20** | 231.20 | 232.80 | 0.16 | 0.16 | 0.82 | **3.64** | 4.18 | 5.03 |
| 30 | 23 | **49.00** | 50.70 | 51.90 | **0.00** | 2.79 | 4.48 | **1.08** | 1.50 | 1.76 |
| 100 | 25 | 2,164.71 | **2,164.13** | 2,168.52 | 0.04 | **0.01** | 0.22 | **10.02** | 12.28 | 19.81 |
| 100 | 50 | 965.37 | 965.26 | 974.28 | 0.03 | 0.02 | 1.01 | **51.68** | 61.09 | 72.86 |
| 100 | 75 | **245.01** | 245.10 | 262.66 | **0.04** | 0.08 | 8.18 | **24.69** | 30.48 | 81.30 |

Table 8: KIP results comparing $NN^l$, $NN^d$, and $NN^n$. Each row is an average over 10 instances, except for $n = 100$, which is an average over 100 instances. All times in seconds.

| # of edges | budget | $NN^l$ | | $NN^u$ | | $GBT^l$ | | $GBT^u$ | | MKKT | | |
|---|---|---|---|---|---|---|---|---|---|---|---|---|
| | | MRE | Time | MRE | Time | MRE | Time | MRE | Time | MRE-5 | MRE-10 | MRE-30 |
| 10 | 0.25 | 2.44 | 2.63 | 0.36 | **0.01** | 1.43 | 3.26 | 0.38 | 0.09 | 5.78 | 0.51 | **0.10** |
| 10 | 0.5 | 0.39 | 2.90 | 1.22 | **0.01** | 1.33 | 3.54 | 2.74 | 0.07 | 6.47 | 2.17 | **0.00** |
| 10 | 0.75 | 0.48 | 2.23 | 1.32 | **0.00** | 0.47 | 2.19 | 0.91 | 0.06 | 5.80 | **0.02** | 0.06 |
| 20 | 0.25 | 3.62 | 5.02 | 2.46 | **0.02** | 1.36 | 5.02 | 1.23 | 0.23 | 7.59 | 4.96 | **0.67** |
| 20 | 0.5 | 1.56 | 4.91 | 4.41 | **0.03** | 0.98 | 5.02 | 2.06 | 0.21 | 6.05 | 2.57 | **0.69** |
| 20 | 0.75 | **0.17** | 3.44 | 1.03 | **0.01** | 0.43 | 4.75 | 2.27 | 0.13 | 8.00 | 0.35 | 0.23 |
| Average | | 1.44 | 3.52 | 1.80 | **0.01** | 1.00 | 3.96 | 1.60 | 0.13 | 6.61 | 1.76 | **0.29** |

Table 9: DNDP results for $\lambda = 0.1$. Each is averaged across 10 instances. $NN^l$ and $GBT^l$ are the learning-based formulations with slack for the lower-level approximation. $NN^u$ and $GBT^u$ are the learning-based formulations for the upper-level approximation.

| $n$ | $k$ | Objective | | Mean Relative Error (%) | | Times | |
|---|---|---|---|---|---|---|---|
| | | $NN^l$ greedy | $NN^l$ no greedy | $NN^l$ greedy | $NN^l$ no greedy | $NN^l$ greedy | $NN^l$ no greedy |
| 18 | 5 | **308.30** | 314.90 | **0.85** | 2.93 | 0.59 | **0.06** |
| 18 | 9 | **145.60** | 150.50 | **1.17** | 4.28 | 0.59 | **0.07** |
| 18 | 14 | **31.00** | 41.60 | **0.00** | 55.04 | 0.22 | **0.05** |
| 20 | 5 | **390.30** | 404.40 | **0.00** | 3.71 | 0.62 | **0.06** |
| 20 | 10 | **165.40** | 172.00 | **0.55** | 4.06 | 0.66 | **0.05** |
| 20 | 15 | **33.40** | 36.50 | **0.00** | 7.31 | 0.32 | **0.06** |
| 22 | 6 | **385.50** | 390.60 | **0.59** | 1.88 | 0.19 | **0.07** |
| 22 | 11 | **163.20** | 170.80 | **0.00** | 4.31 | 0.28 | **0.07** |
| 22 | 17 | **35.20** | 39.10 | **7.91** | 31.93 | 0.24 | **0.06** |
| 25 | 7 | **438.20** | 446.30 | **0.11** | 1.67 | 2.66 | **0.08** |
| 25 | 13 | **194.90** | 197.20 | **0.89** | 2.58 | 2.75 | **0.07** |
| 25 | 19 | **43.30** | 49.10 | **0.00** | 13.53 | 0.48 | **0.07** |
| 28 | 7 | **518.30** | 537.70 | **0.15** | 3.63 | 0.67 | **0.07** |
| 28 | 14 | **224.90** | 225.90 | **0.21** | 0.61 | 2.10 | **0.08** |
| 28 | 21 | **46.70** | 52.10 | **0.00** | 11.85 | 0.45 | **0.08** |
| 30 | 8 | **536.30** | 556.50 | **0.00** | 3.61 | 1.54 | **0.08** |
| 30 | 15 | **231.20** | 233.70 | **0.21** | 1.20 | 3.64 | **0.09** |
| 30 | 23 | **49.00** | 51.30 | **0.00** | 4.97 | 1.08 | **0.08** |
| 100 | 25 | **2,164.71** | 2,473.08 | **0.00** | 14.22 | 10.02 | **0.54** |
| 100 | 50 | **965.37** | 1,062.92 | **0.04** | 10.23 | 51.68 | **0.52** |
| 100 | 75 | **245.01** | 313.44 | **0.00** | 27.62 | 24.69 | **0.53** |

Table 10: KIP results comparing $NN^l$ with and without greedy-based features $NN^d$. Each row averaged over 10 instances, except for $n = 100$, which is an average over 100 instances. All times in seconds.

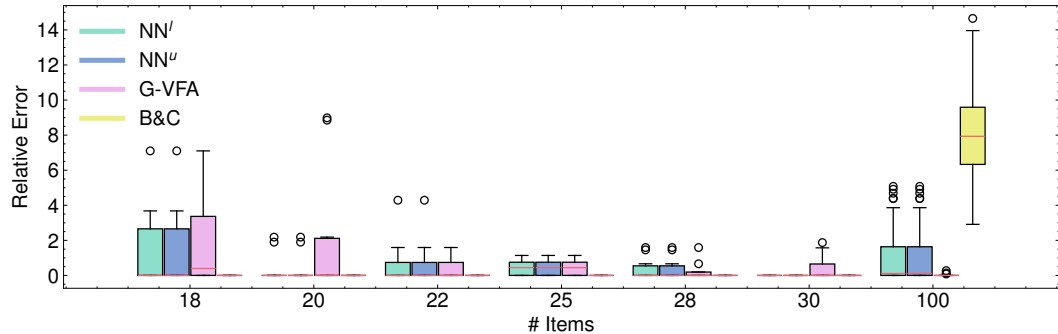

Figure 1: Box plot of relative errors for KIP with interdiction budget of $k = n/4$.

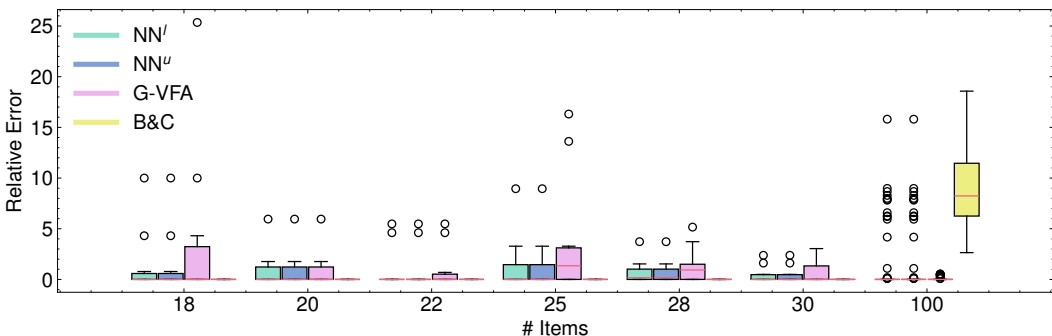

Figure 2: Box plot of relative errors for KIP with interdiction budget of $k = n/2$.

## J Computing Setup

The experiments for the benchmarks were run on a computing cluster with an Intel Xeon CPU E5-2683 and Nvidia Tesla P100 GPU with 64GB of RAM (for training). Pytorch 2.0.1 [44] was used for all neural network models and scikit-learn 1.4.0 was used for gradient-boosted trees in the DNDP [46]. Gurobi 11.0.1 [30] was used as the MILP solver and gurobi-machinelearning 1.4.0 was used to embed the learning models into MILPs.

## K Machine Learning Details

### K.1 Models, features, & hyperparameters

For all problems, we derive features that correspond to each upper-level decision variable, as well as general instance features.

#### K.1.1 KIP, CNP, DRP

For KIP, CNP, DRP, we have $n$ decisions in both the upper- and lower-level of the problems. For the learning model, we utilize a set-based architecture [58], wherein we first represent the objective and constraint coefficients for each upper-level and lower-level decision, independent of the decision ($\mathbf{f}_i$). Each of these are passed through a feed-forward network with shared parameters ($\Psi_d$) to compute an $m$-dimension embedding. The embeddings are then summed and passed through another feed-forward network ($\Psi_s$) to compute the instance's $k$-dimensional embedding. This instance embedding is then concatenated with features related to the upper- and lower-level that are dependent on the decision ($h(\mathbf{x}_i)$). The concatenated vector is passed through a feed-forward network with shared parameters ($\Psi_v$) to predict $n$ scalar values (i.e., one for each decision). The final prediction is equal to the dot product of the $n$ predictions with the objective function coefficients of the upper- or

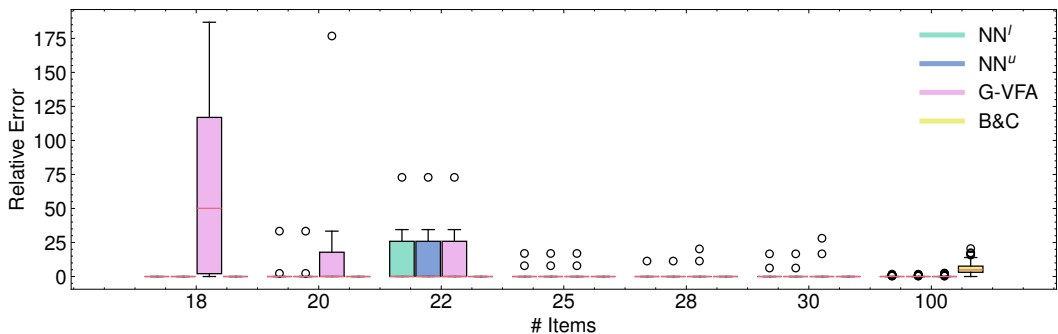

Figure 3: Box plot of relative errors for KIP with interdiction budget of $k = 3n/4$.

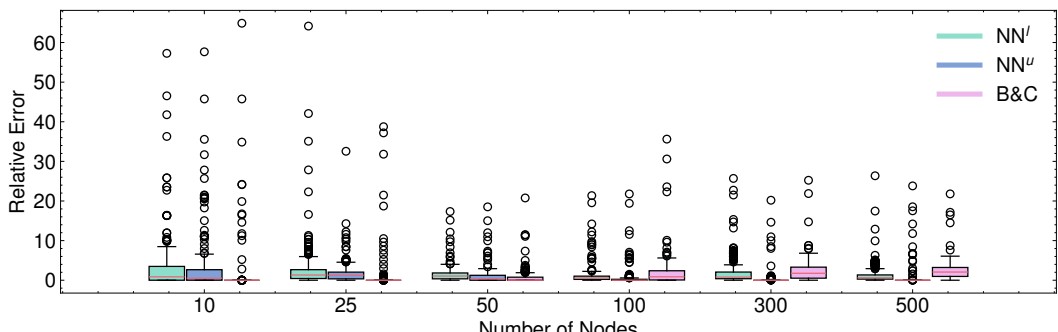

Figure 4: Box plot of relative errors for CNP. B&C does not find any upper-level solutions for 2 of the 300 instances of size $|V| = 500$, so these are excluded from the plot.

lower-level problem, depending on the type of value function approximation. This final step exploits the separable nature of the objective functions in question as they can all be expressed as $\sum_{i=1}^{n} c_i z_i$, where $c_i$ is a *known* coefficient and $z_i$ is a decision variable or a function of a set of decision variables with index $i$. The objectives for KIP, CNP, and DRP all satisfy this property. We leverage this knowledge of the coefficients of separable objective functions as an inductive bias in the design of the learning architecture to facilitate convergence to accurate models. The decision-dependent and decision-independent features are summarized in Table 11.

One minor remark for KIP is that since it is an interdiction problem, we multiply the concatenated vector, i.e., the input to $\Psi_v$, by $(1 - x_i)$ as a mask given that the follower cannot select the same items as the leader.

For all instances, we do not perform systematic hyperparameter tuning. The sub-networks $\Psi_d$, $\Psi_s$, $\Psi_v$ are feed-forward networks with one hidden layer of dimension 128. The decision-independent feature embedding dimension ($m$) is 64, and the instance embedding dimension ($k$) is 32. We use a batch size of 32, a learning rate of 0.01, and Adam [31] as an optimizer.

### K.1.2 DNDP

We train neural network models (one hidden layer, 16 neurons, a learning rate of 0.01 with the Adam optimizer) and gradient-boosted trees (default scikit-learn hyperparameters, except for `n_estimators = 50`). The inputs to these models are 30-dimensional binary vectors representing the subset of links selected by the leader.

### K.2 Data collection & training times

For KIP, CNP, DRP, we sample 1,000 instances according to the procedures specified in Tang et al. [54], Dragotto et al. [18], and Ghatkar et al. [27], respectively. For each instance, we sample 100

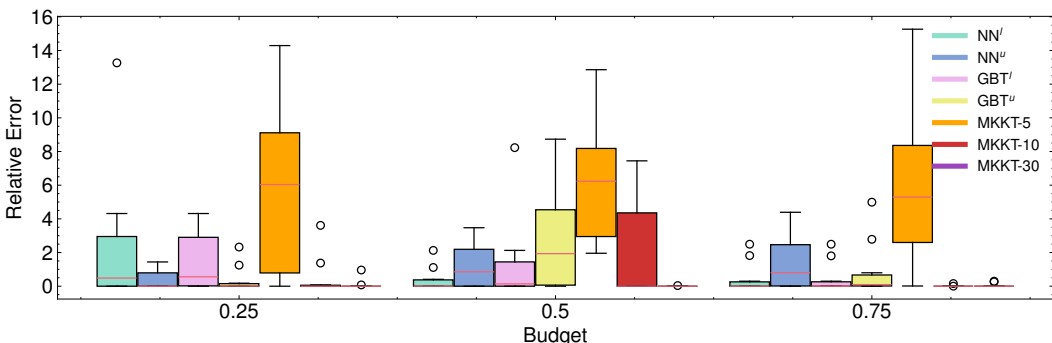

Figure 5: Box plot of relative errors for DNDP with 10 edges. MKKT-{5,10,30} corresponds to MKKT run with each respective time limit.

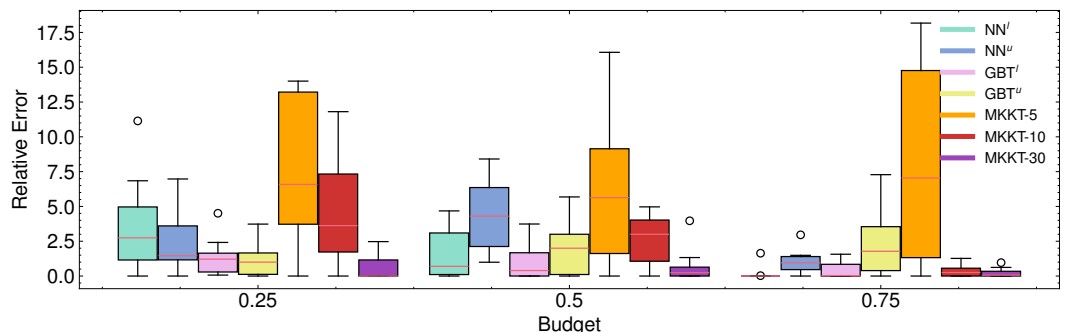

Figure 6: Box plot of relative errors for DNDP with 20 edges. MKKT-{5,10,30} corresponds to MKKT run with each respective time limit.

| Problem | Type | Features |
|---------|------|----------|
| KIP | $\mathbf{f}_i$ | $\frac{p_i/a_i}{\max_i\{p_i/a_i\}}, p_i, a_i, k/n, x_i^{dg}, y_i^{dg}, obj^{dg}/n$ |
|     | $h(\mathbf{x}_i)$ | $\mathbf{f}_i, x_i, y_i^g$ |
| CNP | $\mathbf{f}_i$ | $\frac{p_i^d/d_i}{\max_i\{p_i^d/d_i\}}, \frac{p_i^a/a_i}{\max_i\{p_i^a/a_i\}}, d_i, a_i, p_i^a, p_i^d, \gamma, \eta, \epsilon, \delta, A, D$ |
|     | $h(\mathbf{x}_i)$ | $\mathbf{f}_i, x_i, -\gamma(1-x_i), (1-x_i), (1-\eta)x_i$ |
| DRP | $\mathbf{f}_i$ | $\frac{w_i/c_i}{\max_i\{w_i/c_i\}}, \frac{v_i/c_i}{\max_i\{c_i/v_i\}}, w_i, v_i, c_i, B_d, B_r$ |
|     | $h(\mathbf{x}_i)$ | $\mathbf{f}_i, x_i$ |

Table 11: Features for KIP, CNP, and DRP. Most features are derived directly from the objective and constraint coefficients, so refer to Appendix E for the definitions. For KIP, additional features are computed using simple greedy heuristics. For the KIP DIF, we compute $x_i^{dg}, y_i^{dg}, obj^{dg}$, which correspond to a purely greedy strategy, i.e., the upper-level interdicts the $k$ items with the largest profit to cost ratio ($p_i/a_i$) and the lower-level decisions are the largest remaining highest profit to cost ratio items. For $h(\mathbf{x}_i)$ in KIP, we also include lower-level decisions based on G-VFA ($y_i^g$).

upper-level decisions, i.e., 100,000 samples in total. Additionally, for KIP, CNP, DRP, the lower-level problems are solved with 30 CPUs in parallel. For training, we train for 1,000 epochs. However, if the validation mean absolute error does not improve in 200 iterations, we terminate early. Data collection and training times are reported in Table 12.

For DNDP, we use the Sioux Falls transportation network provided by [47] along with the author's 60 test instances. All instances use the same base network with different sets of candidate links to

add and different budgets. There are 30 candidate links in total, and each test instance involves a subset of 10 or 20 of these links. To construct a training set, we sample 1000 leader decisions by first uniformly sampling an integer between 1 and 20, then uniformly sampling that many candidate links out of the set of 30 options; samples with total cost exceeding 50% of the total cost of all 30 edges are rejected as they are likely to exceed realistic budgets.

| Problem | Data Collection | Training Time Lower | Upper |
|---|---|---|---|
| KIP ($n = 18$) | 142.08 | 2576.43 | - |
| KIP ($n = 20$) | 172.65 | 4714.88 | - |
| KIP ($n = 22$) | 141.61 | 2346.20 | - |
| KIP ($n = 25$) | 170.30 | 4007.75 | - |
| KIP ($n = 28$) | 142.34 | 2684.80 | - |
| KIP ($n = 30$) | 168.91 | 1835.27 | - |
| KIP ($n = 100$) | 164.16 | 3467.26 | - |
| CNP ($|V| = 10$) | 1,397.58 | 1839.60 | 4670.87 |
| CNP ($|V| = 25$) | 1,522.32 | 2072.60 | 4841.31 |
| CNP ($|V| = 50$) | 1,823.16 | 2103.50 | 2963.64 |
| CNP ($|V| = 100$) | 1,872.07 | 1944.08 | 2931.43 |
| CNP ($|V| = 300$) | 3,662.89 | 3800.02 | 3598.04 |
| CNP ($|V| = 500$) | 4,742.06 | 2263.68 | 6214.35 |
| DRP | 1939.24 | 1768.82 | 1784.15 |
| DNDP | 1033.15 | 1.96 | 3.19 |

Table 12: Data collection and training times for all problems. Note that as KIP is an interdiction problem, the same trained model can be used for the upper- and lower-level approximation, so we simply leave the upper-level as - for this problem. All times in seconds.

### K.3 Prediction error

For KIP, CNP, and DRP, we provide the Mean Absolute Error (MAE), as well as the Mean Absolute Label (MAL) as a reference to access the prediction quality for the validation data in Table 13. The table shows that models achieve a MAE of at most $\sim 1e^{-6}$ with a MAL ranging from 0.006 to 200 for all KIP, CNP, and DRP instances. For DNDP, both neural network and gradient-boosted tree models achieve Mean Absolute Percentage Error (MAPE) $\sim 5\%$.

| Problem | Upper-Level Approximation MAE | MAL | Lower-Level Approximation MAE | MAL |
|---|---|---|---|---|
| KIP ($n = 18$) | $5.05e^{-09}$ | 2.0992 | - | - |
| KIP ($n = 20$) | $5.98e^{-09}$ | 2.5369 | - | - |
| KIP ($n = 22$) | $4.00e^{-06}$ | 2.6933 | - | - |
| KIP ($n = 25$) | $3.46e^{-10}$ | 3.0036 | - | - |
| KIP ($n = 28$) | $1.48e^{-08}$ | 3.7327 | - | - |
| KIP ($n = 30$) | $2.85e^{-08}$ | 3.7445 | - | - |
| KIP ($n = 100$) | $1.21e^{-08}$ | 13.104 | - | - |
| CNP ($|V| = 10$) | $1.35e^{-08}$ | 5.4606 | $6.74e^{-06}$ | 1.5272 |
| CNP ($|V| = 25$) | $1.22e^{-06}$ | 12.3452 | $1.09e^{-08}$ | 3.7224 |
| CNP ($|V| = 50$) | $1.23e^{-07}$ | 23.9687 | $4.77e^{-09}$ | 7.7536 |
| CNP ($|V| = 100$) | $4.33e^{-08}$ | 46.5468 | $3.97e^{-06}$ | 14.7491 |
| CNP ($|V| = 300$) | $1.83e^{-08}$ | 135.8972 | $2.54e^{-07}$ | 44.577 |
| CNP ($|V| = 500$) | $8.54e^{-08}$ | 222.3805 | $9.51e^{-08}$ | 76.212 |
| DRP | $2.51e^{-07}$ | 0.0062 | $7.82e^{-08}$ | 0.0703 |
| DNDP | 0.0292 | 0.4297 | 0.02504 | 0.4990 |

Table 13: Prediction errors for all problems. Note that as KIP is an interdiction problem, the same trained model can be used for the upper- and lower-level approximation, so we simply leave the lower-level as - for this problem.

