# OpenReview forum: "Neur2BiLO: Neural Bilevel Optimization"
_NeurIPS.cc/2024/Conference — NeurIPS 2024 poster_

### Official Review · Reviewer_otky · 2024-07-03

**Soundness:** 4
**Presentation:** 4
**Contribution:** 4
**Rating:** 8
**Confidence:** 3

**Summary:**

This paper proposes two approximate methods for solving constrained, mixed-integer, non-linear bilevel optimization problems. The core idea is to convert the bilevel problem into a single level problem by clever use of neural networks trained offline by solving single level optimization problems. The upper-level approach trains a NN to predict for a fixed leader decision, the leader objective function assuming the follower acts optimally. A single level optimization problem can then be solved by replacing the lower level problem with the NN. The other approach uses the value function reformulation for bilevel problems and aims to train a neural network to learn the optimal value of the lower level problem, given a leader decision.

The performance/capability tradeoffs for each method across different classes of bilevel problems is thoroughly discussed, and some theoretical analysis is provided which depends on the approximation error of the true objective/value function. Importantly, the algorithm has a "post-processing" step where the approximate solutions from Neur2BiLO are refined to ensure feasibility of the original bilevel problem.

Neur2BiLO is thoroughly evaluated on several challenging benchmark problems to highlight the generality and effectiveness compared to exact baselines and other learning-based methods.

**Strengths:**

This is a very strong submission overall and was enjoyable to review. In particular, the writing is of extremely high quality given the technical nature of the paper. The methodology is clearly written, but importantly, the discussion of related works, experimental setup, limitations and analysis of results are presented very cleanly.

Neur2BiLO is comprehensively evaluated against exact methods and other learning based methods, both experimentally and in the discussion. The approach is relatively straightforward but seems to integrate machine learning into existing algorithms and results for bilevel optimization a very elegant and principled way. In particular, the practicality of training the method and the effectiveness in practice as shown in the experiments is very appealing.

**Weaknesses:**

This is a minor weakness but none of the problems evaluated have coupled constraints. It would be great if Assumption 1 i) were satisfied but the constraints were coupled.

The theory around the approximation error to the true underlying functions is discussed in Sec. 3.1, which is nice, but it is unclear whether or not for these class of problems that the optimal value function of the follower's problem is actually smooth. I think this is important to at least have some discussion of this to caveat the bounds, since the approximation error in (8) may not be attainable.

**Questions:**

- Lines 161-173: My understanding is that this is a sensible heuristic approach which refines the approximation to satisfy constraints 2b, 2c imposed in (1). Specifically, there is no guarantee that the solution is optimal, but it is probably a good solution which is feasible. Can you please confirm or correct my intuition please? It was not completely clear from reading the paper whether or not this is the case.

- This relates to one of the weaknesses. Is it reasonable to assume in general that the value function of the lower level problem or the optimal objective function of the outer level function is smooth?

**Limitations:**

Limitations have been discussed.

---

> ### Author Rebuttal · Authors · 2024-08-06
>
> Thank you for the review.  Below, we provide responses to each of the weaknesses and questions.
>
> **Weaknesses**:
> - Indeed, none of the problems we study in our experiments contain coupling constraints. We have not found bilevel optimization benchmarks with coupling constraints in the literature. For example, the bilevel mixed-integer linear solver of Fischetti et al. [1] is evaluated on many problems, none of which have coupling as far as we could tell. We will discuss this in the limitations of the paper. Please also note that in the presence of coupling constraints the upper-level approximation algorithm we developed is not applicable since the coupling constraints cannot be modeled.  However, the lower-level approximation can certainly be used.
> - Regarding the smoothness of the optimal value function (OVF): unfortunately even in the simplest possible case, namely when the follower problem is a linear optimization problem with continuous decision variables $y$, the OVF can be non-smooth. Indeed, the OVF of a linear problem is piecewise linear and convex in the right-hand-side parameters of the constraints and piecewise linear and concave in the objective parameters; see [2]. Hence, if the leader variable $x$ appears either on the right-hand-side of the constraints or in the objective function the optimal value function can be piecewise linear and hence non-smooth. Even worse, if the leader variables appear in the constraint matrix of the follower the optimal value function can be discontinuous. Clearly, certain problem structures exist, where the optimal value function of the follower problem is smooth, but we do not see why this would be important for our method. Especially the approximation error in (8) you are mentioning is not affected by the smoothness of the function. Actually, the neural network with ReLU activations is a piecewise linear function itself and thus non-smooth. Hence, at least theoretically, it could achieve an approximation error of $\alpha=0$ in (8). Could you clarify your point about why the smoothness of the OVF would be important for our work?
>
> **Questions**:
> - Yes, you are completely right. There is no guarantee that the solutions returned by our methods are optimal, but in Lines 161-173 we discuss under which assumptions our methods can guarantee a feasible solution. Note that the only case which can lead to an infeasible solution $x^\star$ is if only Assumption 1(ii) is satisfied and if we use the upper-level approximation method. In all other cases the procedure guarantees feasibility. Note also that for the lower level approximation feasibility is always ensured. We will make sure to clarify this point in the paper.
> - Regarding smoothness, please see our answer above.
>
> **References**:
> - [1] Fischetti, M., Ljubić, I., Monaci, M., & Sinnl, M. (2017). A new general-purpose algorithm for mixed-integer bilevel linear programs. Operations Research, 65(6), 1615-1637.
> - [2] Bertsimas, D., & Tsitsiklis, J. N. (1997). Introduction to linear optimization (Vol. 6, pp. 479-530). Belmont, MA: Athena Scientific.

---

> > ### Comment · Reviewer_otky · 2024-08-12
> > **Response to Authors' rebuttal**
> >
> > Thank you for your detailed response and for agreeing to address my concerns and limitations. I still think this is a very strong paper and would advocate for acceptance.
> >
> > Re my concern about smoothness: My question was motivated by the universal approximation theorem and the general ability for a neural network to approximate a function which may not be continuous (the value function in this case). It would be nice to have a little bit of discussion around the approximation error (empirical insights or general discussion like in your response will suffice) for the other more general problems where discontinuous value functions may arise.
> >
> > I appreciate that Theorem 3.1 is for a particular class of problems and does not require any assumption on the value function specifically.

---

> > > ### Author Response · Authors · 2024-08-13
> > > **Response to reviewer**
> > >
> > > Thank you for the clarification.  We agree that including a discussion on this would be a great addition, and we will do so in the final version of the paper.

---

### Official Review · Reviewer_saS9 · 2024-07-07

**Soundness:** 3
**Presentation:** 3
**Contribution:** 3
**Rating:** 6
**Confidence:** 4

**Summary:**

The paper tackles the bi-level optimization problem (BiLo) in general. BiLO can be seen as the problem of a leader computing a strategy (x) to commit to, such that the leader’s objective is optimized subject to the follower’s best response (y) to the committed strategy. The paper provides two ML-based approaches, one is the upper-level approximation that learns to predict a mapping from x to the leader’s objective, and the other is the lower-level approximation that learns to predict the utility of the follower given x. Both approaches reformulate the BiLO into a single-level mathematical program. The paper also provides approximation analyses for the lower-level approximation. The error term is an additive function of ML regression errors and a gap of f values due to discontinuity. In experiments, both approaches are evalutated on 4 different BiLO problems and compared against B&C, heuristics and exact solvers.

**Strengths:**

The paper presents two novel ML-based approaches to reduce the challenging problem of BiLO. The approaches are not complicated but new and interesting. Empirical results show that both methods are promising - they can find similar quality or better quality solutions than the baselines but with shorter runtime.

The paper is also easy to read.

**Weaknesses:**

Note that I have reviewed this paper in the past and I am frankly surprised that it was rejected.
The authors have addressed most of my concerns last time.

These are not major weaknesses but still should be pointed out:
The effectiveness of this approach mainly depends on how closely ML can learn to approximate \Phi(x) or F(x,y*). In general, this looks like a very challenging task and it is the bottleneck of these approaches. In this case, it just happens that the regression task is easy for the four benchmarks.
Furthermore, the theoretical approximation guarantees are not that surprising given that the prediction error is assumed to be bounded.

**Questions:**

I don't have any questions.

**Limitations:**

The applicability of this approach seems to be largely dependent on the ML regression errors. The authors have discussed limitations of their work in the paper.

---

> ### Author Rebuttal · Authors · 2024-08-06
>
> Thank you for the review.
>
> We do acknowledge that the effectiveness certainly does depend on how well the value functions can be approximated.  Through our experiments, we do indeed demonstrate that this is relatively easy for the problems studied and note that these are already challenging problems within bilevel optimization.  While it may be possible that optimization problems with a more complex structure may be harder to approximate, we note that these would pose similar challenges for any method for bilevel optimization.  Specifically, most methods require frequent evaluation of the upper- and/or lower-level problems, the addition of cutting planes, and branch-and-bound.  All of which will likely suffer from similar issues with increasing problem complexity.  Furthermore, more challenging problems are even less likely to have well-defined problem-specific heuristics/algorithms.

---

> > ### Comment · Reviewer_saS9 · 2024-08-12
> >
> > Thanks for the response. I will keep my score.

---

### Official Review · Reviewer_qVCc · 2024-07-11

**Soundness:** 3
**Presentation:** 2
**Contribution:** 2
**Rating:** 4
**Confidence:** 3

**Summary:**

The paper develops a neural method to solve bi-level optimization problems. It begins with a motivating application and then proposes NEUR2BILO, which utilizes two layers of neural networks to approximate solutions for the upper and lower levels.

**Strengths:**

I found the work conducted to be substantial, supported by several proofs of components.

Additionally, the experiments evaluate performance across four scenarios: KIP, CNP, DRP, and DNDP, which adds further substance to the study.

**Weaknesses:**

1. The paper is not very friendly to readers who are not engaged in this specialized area. The problem addressed has many challenging issues, such as bi-level, non-linear, and mixed-variable components. I wonder how the authors address each of these challenges, as it is not very clear in the current presentation.

2. The paper introduces two models, NN^u and NN^l, designed to generate solutions for the upper and lower levels, respectively. Then, I would expect that  the combination of the two models will solve the entire bi-level problem. However, in the experiments, NN^u and NN^l are compared in parallel, which is confusing.

3. I am struggling to identify the major contribution of this study to the learning to optimize area. The paper's organization makes it difficult to grasp the main points clearly.

4. The major body of the paper is not self-contained, requiring frequent switching between the main text and the appendix to obtain necessary information.

**Questions:**

See weaknesses.

**Limitations:**

See weaknesses.

---

> ### Author Rebuttal · Authors · 2024-08-06
>
> Thank you for the review.  Below, we provide responses to each of the weaknesses.
> 1. For the first weakness, we will discuss two separate points.
>    - We acknowledge that this paper focuses on relatively specialized bilevel optimization problems/literature.  We will add some basic references of the area in the introduction of the paper.  However, it is a technical conference and bilevel optimization is indeed a technical topic, so the paper certainly does require some background in the area.  Given the area of submission is Optimization (convex and non-convex, discrete, stochastic, robust) and all other reviewers commented that the clarity of the paper is high, we believe the presentation to be appropriate. However, if you have specific suggestions to improve the readability, please let us know and we can attempt to include them in the final revision.
>    - In terms of the non-linear and mixed-integer bilevel problems, our approach handles all of them similarly as the upper- and lower-level approximations proposed rely only on computing optimal objective values to the decomposed single-level problems for data collection, which can reliably be done with any mixed-integer solver.  The trained models are then utilized directly as discussed in Section 3.1.  For this reason, we view this as a strength of the paper, as standard algorithms for bilevel optimization typically require much stronger assumptions or are limited to specific classes of problems, whereas Neur2BiLO can be deployed quite generally.
> 2. $\text{NN}^u$ and $\text{NN}^l$ are separate approaches, based on the same principle of transforming the bilevel problem into a single-level problem. One does so by learning to approximate the upper-level objective and the other does so by learning to approximate the lower-level objective. Each of the two approaches stand on their own; they cannot be combined.
> 3. The major contribution of this paper is on the development of an efficient general learning-based algorithm for bilevel optimization, through the upper- and lower-level learning based approximations presented. In the final version of the paper, we will make the contributions more explicit.  However, we provide a brief list of major contributions below.
>    - Generality: We propose a learning-based approach for bilevel problems particularly in the presence of integer variables or non-linear constraints/objectives. These are extremely challenging problems for classical optimization methods which require specialization to problem structure or significant computational effort.
>    - Efficiency & Efficacy: Neur2BiLO computes high-quality solutions on a variety of bilevel optimization problems, often within orders of magnitude less time than traditional methods for bilevel optimization.  For larger, and more challenging problems, Neur2BiLO computes best-known solutions, in some cases by large margins, such as the 26% improvement over the state-of-the-art solutions for the donor-recipient problem.
>    - Theoretical Guarantees: We provide theoretical guarantees for solution quality in terms of an additive absolute optimality gap which mainly depends on the prediction accuracy of the regression model.
> 4. Given the pagelimit, we aimed to present the central aspects of our approach in the main body of the paper.  We will aim to improve the readability and inclusion of material within the final version of the paper.  If you have any suggestions on what material you believe would be most beneficial to move from the main paper to the appendix, we would be happy to take that more strongly into consideration.

---

### Official Review · Reviewer_Rwxw · 2024-07-11

**Soundness:** 3
**Presentation:** 3
**Contribution:** 2
**Rating:** 4
**Confidence:** 5

**Summary:**

The paper studied bilevel optimization problems with discrete decision variables. The proposed framework, Neur2BiLO, adopts a learning-based approach to solve such problems, which is based on a trained neural network to approximate the leader's or follower's value functions.

**Strengths:**

The paper studied an important problem, and its overall structure is easy to follow. In recent years, research in bilevel optimization has become increasingly popular.  However, most algorithms are dedicated to bilevel programs with continuous decision variables, like the continuous network design problem in transportation and the hyperparameter optimization problem in machine learning. Yet, bilevel optimization problems with discrete decision variables are equally important and have also found applications in various domains. Hence, the relevance and applicability of tackling such problems are well-justified.

**Weaknesses:**

The proposed algorithm is very straightforward in its development and lacks established performance guarantees. This limitation typically necessitates robust numerical validation. Despite multiple experiments, the evidence provided does not convincingly demonstrate superiority over existing methods.

First, the experiments on the discrete network design problem (DNDP) are conducted on the relatively small Sioux-Falls network. I recommend testing on larger networks, such as the Chicago-Sketch [1], to better assess scalability.

[1] https://github.com/bstabler/TransportationNetworks

Second, a significant application, neural architecture search (NAS), is missing from the study. NAS and DNDP share conceptual similarities: one designs neural networks, and the other transportation networks. Traditionally, NAS often employs continuous relaxation for efficient architecture search using gradient descent. However, given that the proposed algorithm directly manages discrete bilevel optimization, it opens the possibility of exploring NAS without needing such relaxations.

[2] Liu, H., Simonyan, K., & Yang, Y. (2018, September). DARTS: Differentiable Architecture Search. In International Conference on Learning Representations.

**Questions:**

1. Can the proposed algorithm be extended to continuous network design problems? If not, please explain the reasons.

2. Can the proposed algorithm be applied to neural architecture search? If not, please explain the reasons.

**Limitations:**

Please find my comments about the limitations in other parts of the review.

---

> ### Author Rebuttal · Authors · 2024-08-06
>
> Thank you for the review.  Below, we respond to each of the weaknesses and questions.
>
> **Weaknesses**:
> - Next to the performance guarantees we provide in the paper, we believe that we have conducted a thorough numerical validation. In total, we test on 2250 instances with ranging instance sizes on a wide variety of benchmark problems (continuous/integer upper/lower-level problems and non-linearity in both levels). For almost every set of larger instances (i.e., more difficult instances), we achieve better solutions on average over exact/heuristic methods, within multiple orders of magnitude less time.
> - For DNDP, our primary motivation for comparing the Sioux Falls instances is their use, as a publicly available benchmark set, in other bilevel DNDP papers (see [1] and corresponding GitHub repo).  While there may be larger networks at the suggested GitHub repository,  these do not have the bilevel instance parameters required. Namely, the travel speeds and capacities of candidate road segments that are to be considered for addition to the network. The author of [1] did this generation exercise using their domain knowledge for the Sioux Falls network, making it suitable for benchmarking.
> - Regarding neural architecture search (NAS), Neur2BiLO can easily be adapted to this problem (see response to the below question for details).  While this is certainly an important bilevel optimization problem within the machine learning community, this could also be argued for any bilevel optimization problem, such as the over 70 applications listed in [2].  Given our extensive numerical study already evaluates Neur2BiLO on four integer bilevel optimization problems of interest to the bilevel optimization community more broadly, we believe this alone to be a notable contribution and sufficient evaluation.  In addition, we note that the generality and evaluation on four benchmarks with such variable structure is already more than the vast majority of bilevel algorithms evaluate on.  Furthermore, most bilevel algorithms are not even as general as Neur2BiLO given they are not suitable for non-linear problems.
>
> **Questions**:
>
> - Neur2BiLO can be applied to purely continuous bilevel problems. However, these are generally well-solved by formulating the problem as a single-level problem using KKT-conditions or duality theory where applicable, or using first-order gradient methods as in the highly effective approach of BOME [3]. Generally, we focus on more challenging integer bilevel problems, wherein existing methods are intractable or have prohibitively long runtime.
> - Yes, given a bilevel formulation of NAS, it can be applied to this problem.  In this context, one can learn to approximate the value function of any training metric that needs to be optimized.  Data collection can be done via sampling over architectures and training those specific architectures.  As you mentioned, Neur2BiLO may be useful given the discrete nature of NAS.  We would like to thank the reviewer for pointing this out as we believe studying Neur2BiLO and extensions for NAS would certainly be an interesting, and potentially high-impact, direction for future work and contribution.  We believe a contribution such as this would likely warrant an independent paper given the large body of existing methods, and literature, much of which can be leveraged within the Neur2BiLO framework.
>
> **References**
> - [1] Rey, D. (2020). Computational benchmarking of exact methods for the bilevel discrete network design problem. Transportation Research Procedia, 47, 11-18.
> - [2] Dempe, S. (2020). Bilevel optimization: theory, algorithms, applications and a bibliography. Bilevel optimization: advances and next challenges, 581-672.
> - [3] Liu, B., Ye, M., Wright, S., Stone, P., & Liu, Q. (2022). Bome! bilevel optimization made easy: A simple first-order approach. Advances in neural information processing systems, 35, 17248-17262.

---

### Decision · Program_Chairs · 2024-09-25

**Decision:**

Accept (poster)

**Comment:**

The reviews of this paper are mixed, but I propose accepting it. The approach is novel according to the reviewer consensus. The main criticism of this work comes from the angle of solving bilevel problems in the area of neural architecture search, but I believe the application area of this paper to be sufficiently different to warrant its acceptance. The experiments are reasonable and show good performance of the approaches presented.

The authors ought to consider the following changes for the camera ready version as pointed out by one of the reviewers:
- Expand the comparison with related methodologies, e.g.:
	[1]D. Rey, H. Bar-Gera, V. V. Dixit, and S. T. Waller, "A branch-and-price algorithm for the bilevel network maintenance scheduling problem," in Transportation Science, 2019.
  [2] T. Tay and C. Osorio, "A sampling strategy for high-dimensional, simulation-based transportation optimization problems," in Transportation Science, 2024.
  [3] Liu, H., Simonyan, K., & Yang, Y. (2018). DARTS: Differentiable Architecture Search. In International Conference on Learning Representations.